# Arabidopsis RCD1 coordinates chloroplast and mitochondrial functions through interaction with ANAC transcription factors

Alexey Shapiguzov[1,2,3†], Julia P Vainonen[1,2†], Kerri Hunter[1,2‡], Helena Tossavainen[4,5‡], Arjun Tiwari[6‡], Sari Järvi[6‡], Maarit Hellman[5], Fayezeh Aarabi[7], Saleh Alseekh[7,8], Brecht Wybouw[9,10], Katrien Van Der Kelen[9,10§], Lauri Nikkanen[6], Julia Krasensky-Wrzaczek[1,2], Nina Sipari[1,11], Markku Keinänen[12], Esa Tyystjärvi[6], Eevi Rintamäki[6], Bert De Rybel[9,10], Jarkko Salojärvi[1,2#], Frank Van Breusegem[9,10], Alisdair R Fernie[7,8], Mikael Brosché[1,2,13], Perttu Permi[4,5,14], Eva-Mari Aro[6], Michael Wrzaczek[1,2], Jaakko Kangasjärvi[1,2]*

[1]Organismal and Evolutionary Biology Research Programme, Faculty of Biological and Environmental Sciences, University of Helsinki, Helsinki, Finland; [2]Viikki Plant Science Center, University of Helsinki, Helsinki, Finland; [3]Institute of Plant Physiology, Russian Academy of Sciences, Moscow, Russia; [4]Program in Structural Biology and Biophysics, Institute of Biotechnology, University of Helsinki, Helsinki, Finland; [5]Department of Chemistry, Nanoscience Center, University of Jyväskylä, Jyväskylä, Finland; [6]Department of Biochemistry / Molecular Plant Biology, University of Turku, Turku, Finland; [7]Max-Planck Institute for Molecular Plant Physiology, Potsdam, Germany; [8]Center of Plant System Biology and Biotechnology, Plovdiv, Bulgaria; [9]Department of Plant Biotechnology and Bioinformatics, Ghent University, Ghent, Belgium; [10]VIB Center for Plant Systems Biology, Ghent, Belgium; [11]Viikki Metabolomics Unit, Faculty of Biological and Environmental Sciences, University of Helsinki, Helsinki, Finland; [12]Department of Environmental and Biological Sciences, University of Eastern Finland, Joensuu, Finland; [13]Institute of Technology, University of Tartu, Tartu, Estonia; [14]Department of Biological and Environmental Science, Nanoscience Center, University of Jyväskylä, Jyväskylä, Finland

*For correspondence:
Jaakko.Kangasjarvi@helsinki.fi

†These authors contributed equally to this work
‡These authors also contributed equally to this work

Present address: §Sciensano, Brussels, Belgium; #School of Biological Sciences, Nanyang Technological University, Singapore, Singapore

Competing interests: The authors declare that no competing interests exist.

**Abstract** Reactive oxygen species (ROS)-dependent signaling pathways from chloroplasts and mitochondria merge at the nuclear protein RADICAL-INDUCED CELL DEATH1 (RCD1). RCD1 interacts in vivo and suppresses the activity of the transcription factors ANAC013 and ANAC017, which mediate a ROS-related retrograde signal originating from mitochondrial complex III. Inactivation of *RCD1* leads to increased expression of mitochondrial dysfunction stimulon (MDS) genes regulated by ANAC013 and ANAC017. Accumulating MDS gene products, including alternative oxidases (AOXs), affect redox status of the chloroplasts, leading to changes in chloroplast ROS processing and increased protection of photosynthetic apparatus. ROS alter the abundance, thiol redox state and oligomerization of the RCD1 protein in vivo, providing feedback control on its function. RCD1-dependent regulation is linked to chloroplast signaling by 3'-phosphoadenosine 5'-phosphate (PAP). Thus, RCD1 integrates organellar signaling from

chloroplasts and mitochondria to establish transcriptional control over the metabolic processes in both organelles.

DOI: https://doi.org/10.7554/eLife.43284.001

## Introduction

Cells of photosynthesizing eukaryotes are unique in harboring two types of energy organelles, the chloroplasts and the mitochondria, which interact at an operational level by the exchange of metabolites, energy and reducing power (*Noguchi and Yoshida, 2008*; *Cardol et al., 2009*; *Bailleul et al., 2015*). Reducing power flows between the organelles through several pathways, including photorespiration (*Watanabe et al., 2016*), malate shuttles (*Scheibe, 2004*; *Zhao et al., 2018*) and transport of photoassimilate-derived carbon rich metabolites from chloroplasts to mitochondria. At the signaling level, the so-called retrograde signaling pathways originating from the organelles influence the expression of nuclear genes (*de Souza et al., 2017*; *Leister, 2019*; *Waszczak et al., 2018*). These pathways provide feedback communication between the organelles and the gene expression apparatus in the nucleus to adjust expression of genes encoding organelle components in accordance with changes in the developmental stage or environmental conditions.

Reactive oxygen species (ROS), inevitable by-products of aerobic energy metabolism, play pivotal roles in plant organellar signaling from both chloroplasts and mitochondria (*Dietz et al., 2016*; *Noctor et al., 2018*; *Waszczak et al., 2018*). Superoxide anion radical ($O_2^{\cdot-}$) is formed in the organelles by the transfer of electrons from the organellar electron transfer chains (ETCs) to molecular oxygen ($O_2$). In illuminated chloroplasts, superoxide anion formed from $O_2$ reduction by Photosystem I (PSI) is converted to hydrogen peroxide ($H_2O_2$) which is further reduced to water by chloroplastic $H_2O_2$-scavenging systems during the water-water cycle (*Asada, 2006*; *Awad et al., 2015*). Chloroplastic ROS production can be enhanced by application of methyl viologen (MV), a chemical that catalyzes shuttling of electrons from PSI to $O_2$ (*Farrington et al., 1973*). The immediate product of this reaction, $O_2^{\cdot-}$, is not likely to directly mediate organellar signaling; however, $H_2O_2$ is involved in many retrograde signaling pathways (*Leister, 2019*; *Mullineaux et al., 2018*; *Waszczak et al., 2018*). Organellar $H_2O_2$ has been suggested to translocate directly to the nucleus (*Caplan et al., 2015*; *Exposito-Rodriguez et al., 2017*), where it can oxidize thiol groups of specific proteins, thereby converting the ROS signal into thiol redox signals (*Møller and Kristensen, 2004*; *Nietzel et al., 2017*). One recently discovered process affected by chloroplastic $H_2O_2$ is the metabolism of 3'-phosphoadenosine 5'-phosphate (PAP). PAP is a toxic by-product of sulfate metabolism produced when cytoplasmic sulfotransferases (SOTs, e.g., SOT12) transfer a sulfuryl group from PAP-sulfate (PAPS) to various target compounds (*Klein and Papenbrock, 2004*). PAP is transported to chloroplasts where it is detoxified by dephosphorylation to adenosine monophosphate in a reaction catalyzed by the adenosine bisphosphate phosphatase 1, SAL1 (*Quintero et al., 1996*; *Chan et al., 2016*). It has been proposed that oxidation of SAL1 thiols directly or indirectly dependent on chloroplastic $H_2O_2$ inactivates the enzyme, and accumulating PAP may act as a retrograde signal (*Estavillo et al., 2011*; *Chan et al., 2016*; *Crisp et al., 2018*).

ROS are also produced in the mitochondria, for example by complex III at the outer side of the inner mitochondrial membrane (*Cvetkovska et al., 2013*; *Ng et al., 2014*; *Huang et al., 2016*; *Wang et al., 2018*). Blocking electron transfer through complex III by application of the inhibitors antimycin A (AA) or myxothiazol (myx) enhances electron leakage and thus induces the retrograde signal. Two known mediators of this signal are the transcription factors ANAC013 (*De Clercq et al., 2013*) and ANAC017 (*Ng et al., 2013b*; *Van Aken et al., 2016b*) that are both bound to the endoplasmic reticulum (ER) by a transmembrane domain. Mitochondria-derived signals lead to proteolytic cleavage of this domain. The proteins are released from the ER and translocated to the nucleus where they activate the mitochondrial dysfunction stimulon (MDS) genes (*De Clercq et al., 2013*; *Van Aken et al., 2016a*). MDS genes include the mitochondrial alternative oxidases (*AOXs*), *SOT12*, and *ANAC013* itself, which provides positive feedback regulation and thus enhancement of the signal.

Whereas multiple retrograde signaling pathways have been described in detail (*de Souza et al., 2017*; *Leister, 2019*; *Waszczak et al., 2018*), it is still largely unknown how the numerous chloroplast- and mitochondria-derived signals are integrated and processed by the nuclear gene

**eLife digest** Most plant cells contain two types of compartments, the mitochondria and the chloroplasts, which work together to supply the chemical energy required by life processes. Genes located in another part of the cell, the nucleus, encode for the majority of the proteins found in these compartments.

At any given time, the mitochondria and the chloroplasts send specific, 'retrograde' signals to the nucleus to turn on or off the genes they need. For example, mitochondria produce molecules known as reactive oxygen species (ROS) if they are having problems generating energy. These molecules activate several regulatory proteins that move into the nucleus and switch on MDS genes, a set of genes which helps to repair the mitochondria. Chloroplasts also produce ROS that can act as retrograde signals. It is still unclear how the nucleus integrates signals from both chloroplasts and mitochondria to 'decide' which genes to switch on, but a protein called RCD1 may play a role in this process. Indeed, previous studies have found that Arabidopsis plants that lack RCD1 have defects in both their mitochondria and chloroplasts. In these mutant plants, the MDS genes are constantly active and the chloroplasts have problems making ROS.

To investigate this further, Shapiguzov, Vainonen et al. use biochemical and genetic approaches to study RCD1 in Arabidopsis. The experiments confirm that this protein allows a dialog to take place between the retrograde signals of both mitochondria and chloroplasts. On one hand, RCD1 binds to and inhibits the regulatory proteins that usually activate the MDS genes under the control of mitochondria. This explains why, in the absence of RCD1, the MDS genes are always active, which is ultimately disturbing how these compartments work.

On the other hand, RCD1 is also found to be sensitive to the ROS that chloroplasts produce. This means that chloroplasts may be able to affect when mitochondria generate energy by regulating the protein. Finally, further experiments show that MDS genes can affect both mitochondria and chloroplasts: by influencing how these genes are regulated, RCD1 therefore acts on the two types of compartments.

Overall, the work by Shapiguzov, Vainonen et al. describes a new way Arabidopsis coordinates its mitochondria and chloroplasts. Further studies will improve our understanding of how plants regulate these compartments in different environments to produce the energy they need. In practice, this may also help plant breeders create new varieties of crops that produce energy more efficiently and which better resist to stress.

DOI: https://doi.org/10.7554/eLife.43284.002

expression system. Nuclear cyclin-dependent kinase E is implicated in the expression of both chloroplastic (*LHCB2.4*) and mitochondrial (*AOX1a*) components in response to perturbations of chloroplast ETC (*Blanco et al., 2014*), mitochondrial ETC, or $H_2O_2$ treatment (*Ng et al., 2013a*). The transcription factor ABI4 is also suggested to respond to retrograde signals from both organelles (*Giraud et al., 2009*; *Blanco et al., 2014*), although its significance in chloroplast signaling has recently been disputed (*Kacprzak et al., 2019*). Mitochondrial signaling *via* ANAC017 was recently suggested to converge with chloroplast PAP signaling based on similarities in their transcriptomic profiles (*Van Aken and Pogson, 2017*). However, the mechanistic details underlying this convergence remain currently unknown.

Arabidopsis RADICAL-INDUCED CELL DEATH1 (RCD1) is a nuclear protein containing a WWE, a PARP-like [poly (ADP-ribose) polymerase-like], and a C-terminal RST domain (RCD1-SRO1-TAF4) (*Overmyer et al., 2000*; *Ahlfors et al., 2004*; *Jaspers et al., 2009*; *Jaspers et al., 2010a*). In yeast two-hybrid studies RCD1 interacted with several transcription factors (*Jaspers et al., 2009*) including ANAC013, DREB2A (*Vainonen et al., 2012*), and Rap2.4a (*Hiltscher et al., 2014*) *via* the RST domain (*Jaspers et al., 2010b*), and with the sodium transporter SOS1 (*Katiyar-Agarwal et al., 2006*). In agreement with the numerous potential interaction partners of RCD1, the *rcd1* mutant demonstrates pleiotropic phenotypes in diverse stress and developmental responses (*Jaspers et al., 2009*). It has been identified in screens for sensitivity to ozone (*Overmyer et al., 2000*), tolerance to MV (*Fujibe et al., 2004*) and redox imbalance in the chloroplasts (*Heiber et al., 2007*; *Hiltscher et al., 2014*). RCD1 was found to complement the deficiency of the redox sensor YAP1 in

yeast (*Belles-Boix et al., 2000*). Under standard growth conditions, the *rcd1* mutant displays differential expression of over 400 genes, including those encoding mitochondrial AOXs (*Jaspers et al., 2009*; *Brosché et al., 2014*) and the chloroplast 2-Cys peroxiredoxin (2-CP) (*Heiber et al., 2007*; *Hiltscher et al., 2014*).

Here we have addressed the role of RCD1 in the integration of ROS signals emitted by both mitochondria and chloroplasts. Abundance, redox status and oligomerization state of the nuclear-localized RCD1 protein changed in response to ROS generated in the chloroplasts. Furthermore, RCD1 directly interacted in vivo with ANAC013 and ANAC017 and appeared to function as a negative regulator of both transcription factors. The RST domain, mediating RCD1 interaction with ANAC transcription factors, was required for plant sensitivity to chloroplastic ROS. We demonstrate that RCD1 is a molecular component that integrates organellar signal input from both chloroplasts and mitochondria to exert its influence on nuclear gene expression.

## Results

### The response to chloroplastic ROS is compromised in *rcd1*

Methyl viologen (MV) enhances ROS generation in illuminated chloroplasts by catalyzing the transfer of electrons from Photosystem I (PSI) to molecular oxygen. This triggers a chain of reactions that ultimately inhibit Photosystem II (PSII) (*Farrington et al., 1973*; *Nishiyama et al., 2011*). To reveal the significance of nuclear protein RCD1 in these reactions, rosettes of Arabidopsis were pre-treated with MV in darkness. Without exposure to light, the plants displayed unchanged PSII photochemical yield (Fv/Fm). Illumination resulted in a decrease of Fv/Fm in wild type (Col-0), but not in the *rcd1* mutant (*Figure 1A*), suggesting increased tolerance of *rcd1* to chloroplastic ROS production. Analysis of several independent *rcd1* complementation lines expressing different levels of HA-tagged RCD1 revealed that tolerance to MV inversely correlated with the amount of expressed RCD1 (*Figure 1—figure supplements 1* and *2*). This suggests that RCD1 protein quantitatively lowered the resistance of the photosynthetic apparatus to ROS.

Treatment with MV leads to formation of superoxide that is enzymatically dismutated to the more long-lived $H_2O_2$. Chloroplastic production of $H_2O_2$ in the presence of MV was assessed by staining plants with 3,3′-diaminobenzidine (DAB) in light. Higher production rate of $H_2O_2$ was evident in MV pre-treated rosettes of both Col-0 and *rcd1*. Longer illumination led to a time-dependent increase in the DAB staining intensity in Col-0, but not in *rcd1* (*Figure 1—figure supplement 3*). In several MV-tolerant mutants, the resistance is based on restricted access of MV to chloroplasts (*Hawkes, 2014*). However, in *rcd1* MV pre-treatment led to an initial increase in $H_2O_2$ production rate similar to that in the wild type (*Figure 1—figure supplement 3*), suggesting that resistance of *rcd1* was not due to lowered delivery of MV to PSI. To test this directly, the kinetics of PSI oxidation was assessed by in vivo spectroscopy using DUAL-PAM. As expected, pre-treatment of leaves with MV led to accelerated oxidation of PSI. This effect was identical in Col-0 and *rcd1*, indicating unrestricted access of MV to PSI in the *rcd1* mutant (*Figure 1B*).

The MV toxicity was not associated with the changed stoichiometry of photosystems (*Figure 1—figure supplement 4A*). However, in Col-0 it coincided with progressive destabilization of PSII complex with its light-harvesting antennae (LHCII) and accumulation of PSII monomer (*Figure 1—figure supplement 4B*). No signs of PSI inhibition were evident either in DUAL-PAM (*Figure 1B*) or in PSI immunoblotting assays (*Figure 1—figure supplement 4B*) in either genotype. The fact that production of ROS affected PSII, but not PSI where these ROS are formed, suggests that PSII inhibition results from a regulated mechanism rather than uncontrolled oxidation by ROS, and that this mechanism requires the activity of RCD1.

Previous studies have described *rcd1* as a mutant with altered ROS metabolism and redox status of the chloroplasts, although the underlying mechanisms are unknown (*Fujibe et al., 2004*; *Heiber et al., 2007*; *Hiltscher et al., 2014*; *Cui et al., 2019*). No significant changes were detected in *rcd1* in transcript levels of chloroplast-related genes (*Brosché et al., 2014*). Analyses of the low molecular weight antioxidant compounds ascorbate and glutathione did not explain the tolerance of *rcd1* to chloroplastic ROS either (*Heiber et al., 2007*; *Hiltscher et al., 2014*). To understand the molecular basis of the RCD1-dependent redox alterations, the levels of chloroplast proteins related to photosynthesis and ROS scavenging were analyzed by immunoblotting. None of these showed

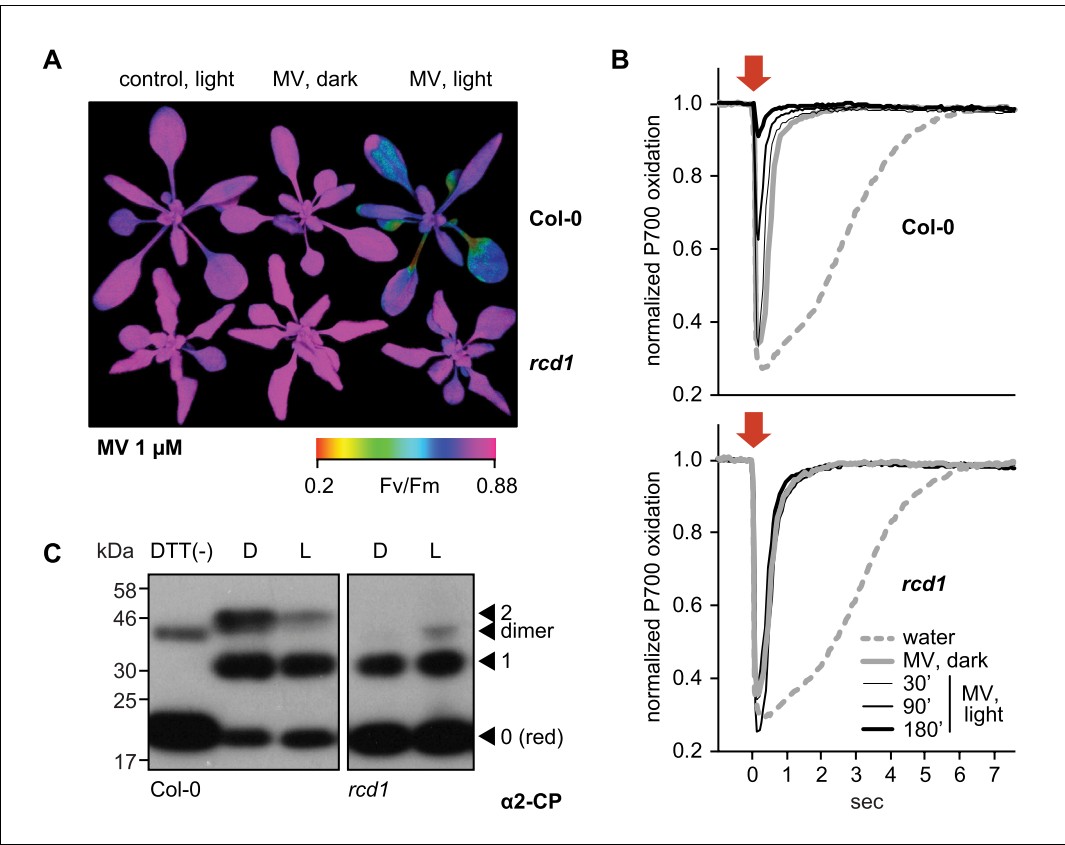

**Figure 1.** RCD1 controls tolerance of photosynthetic apparatus to ROS. (**A**) MV treatment results in PSII inhibition under light, which is suppressed in the *rcd1* mutant. PSII photochemical yield (Fv/Fm) was measured in rosettes pre-treated overnight in darkness with 1 μM MV and then exposed to 3 hr of continuous light (80 μmol m$^{-2}$ s$^{-1}$). Representative false-color image of Fv/Fm is shown. (**B**) Access of MV to electron-acceptor side of PSI is unaltered in *rcd1*. Treatment with MV led to similar changes in kinetics of PSI oxidation in Col-0 and *rcd1*. Oxidation of PSI reaction center (P700) was measured using DUAL-PAM. Leaves were first adapted to far-red light that is more efficiently used by PSI than PSII. In these conditions PSI is producing electrons at a faster rate than they are supplied by PSII, thus P700 is oxidized. Then a flash of orange light was provided that is efficiently absorbed by PSII (orange arrow). Electrons generated by PSII transiently reduced PSI, after which the kinetics of PSI re-oxidation was followed. Note the progressive decrease in the effect of the orange flash occurring in Col-0 at later time points, which suggests deterioration in PSII function. This was not observed in *rcd1*. Three leaves from three individual plants were used for each measurement. The experiment was repeated three times with similar results. (**C**) Redox state of the chloroplast enzyme 2-Cys peroxiredoxin (2-CP) assessed by thiol bond-specific labeling in Col-0 (left) and *rcd1* (right). Total protein was isolated from leaves incubated in darkness (**D**), or under light (**L**). Free sulfhydryls were blocked with N-ethylmaleimide, then in vivo thiol bridges were reduced with DTT, and finally the newly exposed sulfhydryls were labeled with methoxypolyethylene glycol maleimide of molecular weight 5 kDa. The labeled protein extracts were separated by SDS-PAGE and immunoblotted with α2-CP antibody. DTT (-) control contained predominantly unlabeled form. Unlabeled reduced (red), singly and doubly labeled oxidized forms and the putative dimer were annotated as in *Nikkanen et al. (2016)*. Apparent molecular weight increment after the labeling of one thiol bond appears on SDS-PAGE higher than 10 kDa because of steric hindrance exerted on branched polymers during gel separation (*van Leeuwen et al., 2017*). The experiment was repeated three times with similar results.

DOI: https://doi.org/10.7554/eLife.43284.003

The following source data and figure supplements are available for figure 1:

**Source data 1.** Source data and statistics.
DOI: https://doi.org/10.7554/eLife.43284.009
**Figure supplement 1.** Inverse correlation of RCD1 abundance with tolerance to chloroplastic ROS.
DOI: https://doi.org/10.7554/eLife.43284.004
**Figure supplement 2.** The Imaging PAM protocol developed to monitor kinetics of PSII inhibition by repetitive 1 hr light cycles.

*Figure 1 continued*

DOI: https://doi.org/10.7554/eLife.43284.005
**Figure supplement 3.** Production rate of hydrogen peroxide in Col-0 and *rcd1* during illumination of MV-pretreated rosettes.
DOI: https://doi.org/10.7554/eLife.43284.006
**Figure supplement 4.** Altered resistance of *rcd1* photosynthetic apparatus to chloroplastic ROS.
DOI: https://doi.org/10.7554/eLife.43284.007
**Figure supplement 5.** Components of photosynthetic electron transfer and chloroplast ROS scavenging; abundance and distribution of $NAD^+$/NADH and $NADP^+$/NADPH redox couples in Col-0 and *rcd1*.
DOI: https://doi.org/10.7554/eLife.43284.008

significantly altered abundance in *rcd1* compared to Col-0 (*Figure 1—figure supplement 5A*). Furthermore, no difference was detected between the genotypes in abundance and subcellular distribution of the nucleotide redox couples $NAD^+$/NADH and $NADP^+$/NADPH (*Figure 1—figure supplement 5B,C*). Finally, the redox status of chloroplast thiol redox enzymes was addressed. The chloroplast stroma-localized 2-Cys peroxiredoxin (2-CP) is an abundant enzyme (*König et al., 2002*; *Peltier et al., 2006*; *Liebthal et al., 2018*) that was recently found to link chloroplast thiol redox system to ROS (*Ojeda et al., 2018*; *Vaseghi et al., 2018*; *Yoshida et al., 2018*). The level of the 2-CP protein was unchanged in *rcd1* (*Figure 1—figure supplement 5A*). However, when protein extracts were subjected to thiol bond-specific labeling (*Nikkanen et al., 2016*) as described in *Figure 1C*, most 2-CP was reduced in *rcd1* both in darkness and in light, while in Col-0 the larger fraction of 2-CP was present as oxidized forms. Thus, RCD1 is likely involved in the regulation of the redox status of chloroplastic thiol enzymes.

Taken together, the results hinted that the mechanisms by which RCD1 regulates chloroplastic redox status are independent of the photosynthetic ETC, or steady-state levels and distribution of nucleotide electron carriers. However, they appear to be associated with changed thiol redox state of chloroplast enzymes.

## RCD1 protein is sensitive to ROS

It was next tested whether the nuclear RCD1 protein could itself be sensitive to ROS, thus accounting for the observed alterations. For that, an RCD1-HA complementation line was used (line '*a*' in *Figure 1—figure supplement 1*). No changes were detected in RCD1-HA abundance during 5 hr amid the standard growth light period, or during 5 hr high light treatment. On the other hand, both MV and $H_2O_2$ treatments led to a gradual decrease in RCD1 abundance (*Figure 2A*). When plant extracts from these experiments were separated in non-reducing SDS-PAGE, the RCD1-HA signal resolved into species of different molecular weights (*Figure 2B*). Under standard growth conditions or high light, most RCD1-HA formed a reduced monomer. In contrast, treatment with MV under light or $H_2O_2$ resulted in fast conversion of RCD1-HA monomers into high-molecular-weight aggregates (*Figure 2B*). Importantly, MV-induced redox changes in RCD1-HA only occurred in light, but not in darkness, suggesting that the changes were mediated by increased chloroplastic ROS production (*Figure 2B* and *Figure 4—figure supplement 2B*). To test whether oligomerization of RCD1 was thiol-regulated, a variant of RCD1-HA was generated where seven cysteines in the linkers between the RCD1 domains were substituted by alanines (RCD1Δ7Cys; *Figure 2—figure supplement 1A*). The treatments of *rcd1*: RCD1Δ7Cys-HA plants with MV or $H_2O_2$ led to significantly less aggregation of RCD1Δ7Cys-HA compared to RCD1-HA. In addition, the levels of RCD1Δ7Cys-HA were insensitive to MV or $H_2O_2$ (*Figure 2—figure supplement 1B*). In three independent complementation lines the RCD1Δ7Cys-HA variant accumulated to higher levels compared to RCD1-HA (*Figure 2—figure supplement 1C*). This suggests the involvement of the tested RCD1 cysteine residues in the regulation of the protein oligomerization and stability in vivo. However, the tolerance of the RCD1Δ7Cys-HA lines to chloroplastic ROS and the expression of the selected RCD1-regulated genes in response to MV treatment were comparable to that of the RCD1-HA lines or Col-0 (*Figure 2—figure supplement 1C,D*). These results suggest that the RCD1 protein is sensitive to chloroplastic ROS. However, the changes in RCD1 abundance and redox state did not explain the RCD1-dependent redox alterations observed in the chloroplasts.

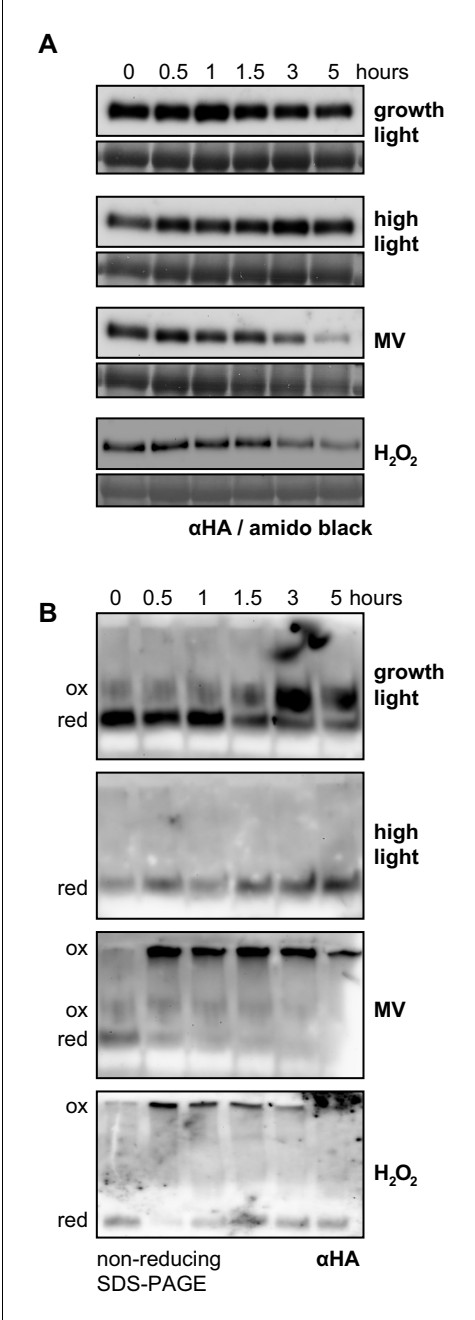

**Figure 2.** RCD1 protein is sensitive to chloroplastic ROS. (**A**) The *rcd1*: RCD1-HA complementation line was used to assess RCD1-HA abundance. It gradually decreased in response to chloroplastic ROS. Leaf discs from plants expressing HA-tagged RCD1 were treated with 5 hr growth light (150 µmol m$^{-2}$ s$^{-1}$), high light (1300 µmol m$^{-2}$ s$^{-1}$), MV (1 µM) in light, or H$_2$O$_2$ (100 mM). The levels of RCD1-HA were monitored by immunoblotting with αHA at indicated time points. Rubisco large subunit (RbcL) detected by amido black staining is shown as a control for equal protein loading. The '0' time point of the MV time course represents dark-adapted leaf discs pre-treated with MV overnight.
*Figure 2 continued on next page*

## Mitochondrial respiration is altered in *rcd1*

In further search for the mechanisms of RCD1-dependent redox alterations in the chloroplast (**Figure 1**), analysis of cell energy metabolism was performed by feeding uniformly labeled [U-$^{14}$C] glucose to leaf discs from light- and dark-adapted Col-0 and *rcd1* plants. Distribution of radioactive label between emitted $^{14}$CO$_2$ and fractionated plant material was analyzed. This revealed significantly more active carbohydrate metabolism in *rcd1* (**Figure 3—source data 1**). The redistribution of radiolabel to sucrose, starch and cell wall was elevated in *rcd1* as were the corresponding deduced fluxes (**Figure 3**), suggesting that *rcd1* displayed a higher respiration rate indicative of mitochondrial defects.

Indeed, earlier transcriptomic studies in *rcd1* have revealed increased expression of genes encoding mitochondrial functions, including mitochondrial alternative oxidases (*AOXs*) (**Jaspers et al., 2009**; **Brosché et al., 2014**). Immunoblotting of protein extracts from isolated mitochondria with an antibody recognizing all five isoforms of Arabidopsis AOX confirmed the increased abundance of AOX in *rcd1* (**Figure 4A**). The most abundant AOX isoform in Arabidopsis is AOX1a. Accordingly, only a weak signal was detected in the *aox1a* mutant. However, in the *rcd1 aox1a* double mutant AOXs other than AOX1a were evident, thus the absence of RCD1 led to an increased abundance of several AOX isoforms.

To test whether the high abundance of AOXs in *rcd1* correlated with their increased activity, seedling respiration was assayed in vivo. Mitochondrial AOXs form an alternative respiratory pathway to the KCN-sensitive electron transfer through complex III and cytochrome c (**Figure 4B**). Thus, after recording the initial rate of O$_2$ uptake, KCN was added to inhibit cytochrome-dependent respiration. In Col-0 seedlings KCN led to approximately 80% decrease in O$_2$ uptake, *versus* only about 20% in *rcd1*, revealing elevated AOX capacity of the mutant (**Figure 4C**). The elevated AOX capacity of *rcd1* was similar to that of an *AOX1a*-OE overexpressor line (**Umbach et al., 2005**). In the *rcd1 aox1a* double mutant the AOX capacity was comparable to Col-0 or *aox1a* (**Figure 4C**). Thus, elevated AOX respiration of *rcd1* seedlings was dependent on the AOX1a isoform. Importantly, however, metabolism of *rcd1 aox1a* was only slightly different from *rcd1* under light and indistinguishable from *rcd1* in the darkness (**Figure 3—source**

*Figure 2 continued*

The experiment was performed four times with similar results. (B) Chloroplastic ROS caused oligomerization of RCD1-HA. Total protein extracts from the plants treated as in panel (A) were separated by non-reducing PAGE and immunoblotted with αHA antibody. Reduced (red) and oxidized (ox) forms of the protein are labeled. To ascertain that all HA-tagged protein including that forming high-molecular-weight aggregates has been detected by immunoblotting, the transfer to a membrane was performed using the entire SDS-PAGE gel including the stacking gel and the well pockets. The experiment was performed four times with similar results.

DOI: https://doi.org/10.7554/eLife.43284.010

The following source data and figure supplement are available for figure 2:

**Source data 1.** Source data and statistics.

DOI: https://doi.org/10.7554/eLife.43284.012

**Figure supplement 1.** Characterization of the *rcd1*: RCD1Δ7Cys-HA lines.

DOI: https://doi.org/10.7554/eLife.43284.011

data 1). This again indicated that the studied phenotypes of *rcd1* are associated with the induction of more than one AOX isoform. Taken together, the results suggested that inactivation of *RCD1* led to increased expression and activity of AOX isoforms, which could contribute to the observed changes in energy metabolism of *rcd1* (*Figure 3*).

## Mitochondrial AOXs affect ROS processing in the chloroplasts

Inhibition of complex III by antimycin A (AA) or myxothiazol (myx) activates mitochondrial retrograde signaling (*Figure 4B*). It leads to nuclear transcriptional reprogramming including induction of *AOX* genes (*Clifton et al., 2006*). Accordingly, overnight treatment with either of these chemicals significantly increased the abundance of AOXs in Col-0, *rcd1* and *rcd1 aox1a* (*Figure 4—figure supplement 1*). Thus, sensitivity of *rcd1* to the complex III retrograde signal was not compromised, rather continuously augmented. In addition, no major effect was observed on RCD1-HA protein level or redox state in the RCD1-HA line treated with AA or myx, suggesting that RCD1 acts as a modulator, not as a mediator, of the mitochondrial retrograde signal (*Figure 4—figure supplement 2*).

To assess whether increased AOX abundance affected chloroplast functions, PSII inhibition was assayed in the presence of MV in AA- or myx-pre-treated leaf discs. Pre-treatment of Col-0 with either AA or myx increased the resistance of PSII to inhibition by chloroplastic ROS (*Figure 4D*), thus mimicking the *rcd1* phenotype. In addition to complex III, AA has been reported to inhibit plastid cyclic electron flow dependent on PGR5 (PROTON GRADIENT REGULATION 5). Thus, *pgr5* mutant was tested for its tolerance to chloroplastic ROS after AA pre-treatment. AA made *pgr5* more MV-tolerant similarly to the wild type, indicating that PGR5 is not involved in the observed gain in ROS tolerance (*Figure 4—figure supplement 3A*).

Mitochondrial complex III signaling induces expression of several genes other than *AOX*. To test whether accumulation of AOXs contributed to PSII protection from chloroplastic ROS or merely correlated with it, the AOX inhibitor salicylhydroxamic acid (SHAM) was used. Treatment of plants with SHAM alone resulted in very mild PSII inhibition, which was similar in *rcd1* and Col-0 (*Figure 4—figure supplement 3B*). However, pre-treatment with SHAM made both *rcd1* and Col-0 plants significantly more sensitive to chloroplastic ROS generated by MV (*Figure 4E*), thereby partially abolishing MV tolerance of the *rcd1* mutant. Involvement of the plastid terminal oxidase PTOX (*Fu et al., 2012*) in this effect was excluded by using the *ptox* mutant (*Figure 4—figure supplement 3C*). Noteworthy, analyses of *AOX1a*-OE, *aox1a* and *rcd1 aox1a* lines demonstrated that AOX1a isoform was neither sufficient nor necessary for chloroplast ROS tolerance (*Figure 4—figure supplement 4*). Taken together, these results indicated that mitochondrial AOXs contributed to resistance of PSII to chloroplastic ROS. We hypothesize that AOX isoforms other than AOX1a are implicated in this process.

## Evidence for altered electron transfer between chloroplasts and mitochondria in *rcd1*

The pathway linking mitochondrial AOXs with chloroplastic ROS processing is likely to involve electron transfer between the two organelles. Chlorophyll fluorescence under light (Fs; *Figure 1—figure supplement 2*) inversely correlates with the rate of electron transfer from PSII to plastoquinone and thus can be used as a proxy of the reduction state of the chloroplast ETC. After combined treatment with SHAM and MV (as in *Figure 4E*), Fs increased in *rcd1*, but not in Col-0 (*Figure 5A*). This hinted that a pathway in *rcd1* linked the chloroplast ETC to the activity of mitochondrial AOXs, with the

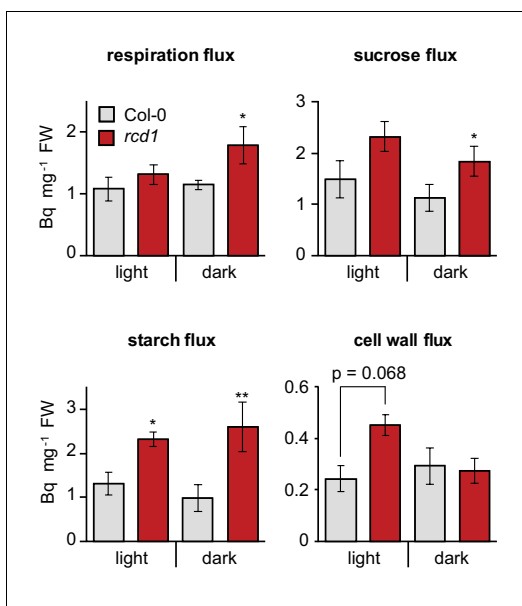

**Figure 3.** Altered energy metabolism of *rcd1*. Deduced metabolic fluxes in light- and dark- adapted Col-0 and *rcd1* rosettes were assessed by fractionation of the extracts of leaves treated with [U-$^{14}$C] glucose. Increased respiration flux and higher amount of total metabolized glucose (*Figure 3—source data 1*) in *rcd1* suggest a more active glycolytic pathway. Higher cell wall metabolic flux in *rcd1* provided indirect support of increased operation of the oxidative pentose phosphate pathway which is required for generating pentoses used in cell wall biosynthesis (*Ap Rees, 1978*). Mean ±SE are presented. Asterisks indicate values significantly different from the wild type, **P value < 0.01, *P value < 0.05, Student's t-test. Source data and statistics are presented in *Figure 3—source data 2*.

DOI: https://doi.org/10.7554/eLife.43284.013

The following source data is available for figure 3:

**Source data 1.** Metabolic analyses.
DOI: https://doi.org/10.7554/eLife.43284.014
**Source data 2.** Source data and statistics.
DOI: https://doi.org/10.7554/eLife.43284.015

latter functioning as an electron sink. When the AOX activity was inhibited by SHAM, electron flow along this pathway was blocked. This led to accumulation of electrons in the chloroplast ETC and hence to the observed rise in Fs. As a parallel approach, dynamics of PSII photochemical quenching was evaluated in MV-pre-treated Col-0 and *rcd1*. In both lines, this parameter dropped within the first 20 min upon exposure to light and then started to recover. Recovery was more pronounced and more suppressed by SHAM in *rcd1* (*Figure 5—figure supplement 1*). These experiments suggest that exposure of MV-pretreated plants to light triggered an adjustment of electron flows, which was compromised by SHAM. This was in line with the involvement of AOXs in photosynthetic electron transfer and chloroplast ROS maintenance.

One of the mediators of electron transfer between the organelles is the malate shuttle (*Scheibe, 2004*; *Zhao et al., 2018*). Thus, malate concentrations were measured in total extracts from Col-0 and *rcd1* seedlings. Illumination of seedlings pre-treated with MV led to dramatic decrease in malate concentration in Col-0, but not in *rcd1* (*Figure 5B*). Noteworthy, under standard light-adapted growth conditions, the concentration and the subcellular distribution of malate was unchanged in *rcd1* (*Figure 5—figure supplement 2*). These observations suggest that exposure to light of MV-pre-treated plants resulted in rearrangements of electron flows that were different in Col-0 and *rcd1*.

Next, the activity of another component of the malate shuttle, the NADPH-dependent malate dehydrogenase (NADPH-MDH), was measured. Chloroplast NADPH-MDH is a redox-regulated enzyme activated by reduction of thiol bridges. Thus, the initial NADPH-MDH activity may reflect the in vivo thiol redox state of the cellular compartment from which it has been isolated. After measuring this parameter, thiol reductant was added to the extracts to reveal the total NADPH-MDH activity. Both values were higher in *rcd1* than in Col-0 (*Figure 5C*). To determine the contribution of in vivo thiol redox state, the initial NADPH-MDH activity was divided by the total activity. This value, the activation state, was also increased in *rcd1* (*Figure 5C*).

Taken together, our results suggested that mitochondria contributed to ROS processing in the chloroplasts *via* a mechanism involving mitochondrial AOXs and possibly the malate shuttle. These processes appeared to be dynamically regulated in response to chloroplastic ROS production, and RCD1 was involved in this regulation.

## Retrograde signaling from both chloroplasts and mitochondria is altered in *rcd1*

Our results demonstrated that absence of RCD1 caused physiological alterations in both chloroplasts and mitochondria. As RCD1 is a nuclear-localized transcriptional co-regulator (*Jaspers et al., 2009*; *Jaspers et al., 2010a*), its involvement in retrograde signaling pathways from both organelles was

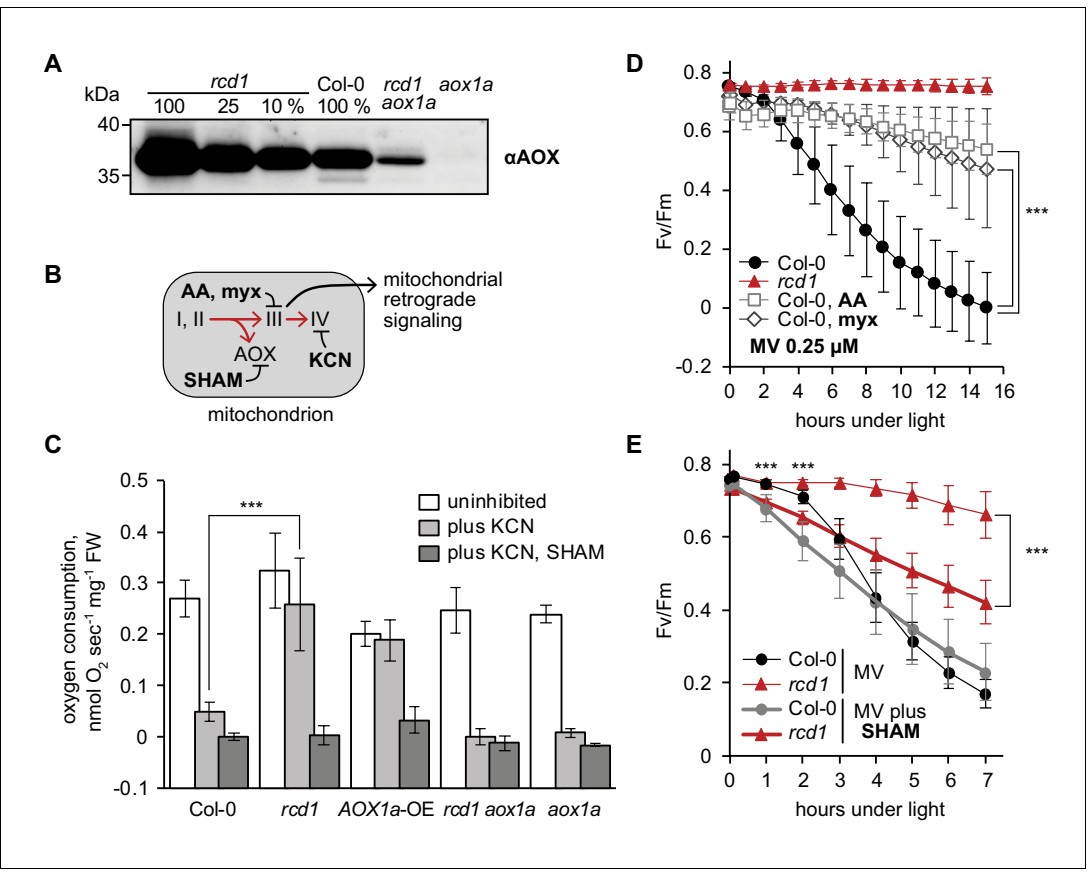

**Figure 4.** Mitochondrial AOXs affect energy metabolism of *rcd1* and alter response to chloroplastic ROS. Source data and statistics are presented in ***Figure 4—source data 1***. (**A**) Expression of AOXs is induced in *rcd1*. Abundance of AOX isoforms in mitochondrial preparations was assessed by immunoblotting with αAOX antibody that recognizes AOX1a, -b, -c, -d, and AOX2 isoforms. 100% corresponds to 15 μg of mitochondrial protein. (**B**) Two mitochondrial respiratory pathways (red arrows) and sites of action of mitochondrial inhibitors. KCN inhibits complex IV (cytochrome c oxidase). Salicylhydroxamic acid (SHAM) inhibits AOX activity. Antimycin A (AA) and myxothiazol (myx) block electron transfer through complex III (ubiquinol-cytochrome c oxidoreductase), creating ROS-related mitochondrial retrograde signal. (**C**) AOX capacity is significantly increased in *rcd1*. Oxygen uptake by seedlings was measured in the darkness in presence of KCN and SHAM. Addition of KCN blocked respiration through complex IV, thus revealing the capacity of the alternative respiratory pathway through AOXs. Data are presented as mean ±SD, asterisks denote selected values that are significantly different (P value < 0.001, one-way ANOVA with Bonferroni post hoc correction). Each measurement was performed on 10–15 pooled seedlings and repeated at least three times. (**D**) Inhibitors of mitochondrial complex III increase plant tolerance to chloroplastic ROS. Effect of pre-treatment with 2.5 μM AA or 2.5 μM myx on PSII inhibition (Fv/Fm) by MV. For each experiment, leaf discs from at least four individual rosettes were used. The experiment was performed four times with similar results. Mean ±SD are shown. Asterisks indicate selected treatments that are significantly different (P value < 0.001, Bonferroni post hoc correction). AOX abundance in the leaf discs treated in the same way was quantified by immunoblotting (***Figure 4—figure supplement 1***). (**E**) AOX inhibitor SHAM decreases plant tolerance to chloroplastic ROS. 1 hr pre-treatment with 2 mM SHAM inhibited tolerance to 1 μM MV both in Col-0 and *rcd1* as measured by Fv/Fm. SHAM stock solution was prepared in DMSO, thus pure DMSO was added in the SHAM-minus controls. For each experiment, leaf discs from at least four individual rosettes were used. The experiment was performed four times with similar results. Mean ±SD are shown. Asterisks indicate significant difference in the treatments of the same genotype at the selected time points (P value < 0.001, Bonferroni post hoc correction).

DOI: https://doi.org/10.7554/eLife.43284.016

The following source data and figure supplements are available for figure 4:

**Source data 1.** Source data and statistics.
DOI: https://doi.org/10.7554/eLife.43284.021

**Figure supplement 1.** Effect of mitochondrial complex III inhibitors on expression of AOXs in Col-0 and *rcd1*.
DOI: https://doi.org/10.7554/eLife.43284.017

*Figure 4 continued on next page*

*Figure 4 continued*

**Figure supplement 2.** Effect of mitochondrial complex III inhibitors on abundance and redox state of the RCD1 protein.
DOI: https://doi.org/10.7554/eLife.43284.018
**Figure supplement 3.** Specificity of inhibitor treatments.
DOI: https://doi.org/10.7554/eLife.43284.019
**Figure supplement 4.** Irrelevance of AOX1a isoform for MV tolerance.
DOI: https://doi.org/10.7554/eLife.43284.020

assessed. Transcriptional changes observed in *rcd1* (*Jaspers et al., 2009*; *Brosché et al., 2014*) were compared to gene expression datasets obtained after perturbations in energy organelles. This revealed a striking similarity of genes differentially regulated in *rcd1* to those affected by disturbed organellar function (*Figure 6—figure supplement 1*). Analyzed perturbations included disruptions of mitochondrial genome stability (*msh1 recA3*), organelle translation (*mterf6*, *prors1*), activity of mitochondrial complex I (*ndufs4*, rotenone), complex III (AA), and ATP synthase function (oligomycin), as well as treatments and mutants related to chloroplastic ROS production (high light, MV, $H_2O_2$, *alx8/fry1*, norflurazon).

In particular, a significant overlap was observed between genes mis-regulated in *rcd1* and the mitochondrial dysfunction stimulon (MDS) genes (*De Clercq et al., 2013*) (*Figure 6A*). Consistently, *AOX1a* was among the genes induced by the majority of the treatments. To address the role of RCD1 protein in the induction of other MDS genes, mRNA steady state levels for some of them were assayed 3 hr after AA treatment (*Figure 6—figure supplement 2*). As expected, expression of all these genes was elevated in *rcd1* under control conditions. Treatment with AA induced accumulation of MDS transcripts to similar levels in Col-0, *rcd1*, and in *rcd1*: RCD1-HA lines that expressed low levels of RCD1. For one marker gene, *UPOX* (*UP-REGULATED BY OXIDATIVE STRESS*), AA induction was impaired in the lines expressing high levels of RCD1-HA or RCD1Δ7Cys-HA (*Figure 6—figure supplement 2*).

In addition to MDS, the list of genes mis-regulated in *rcd1* overlapped with those affected by 3'-phosphoadenosine 5'-phosphate (PAP) signaling (*Estavillo et al., 2011*; *Van Aken and Pogson, 2017*) (*Figure 6A*). Given that PAP signaling is suppressed by the activity of SAL1, expression of PAP-regulated genes was increased in the mutants deficient in SAL1 (*alx8* and *fry1*, *Figure 6A* and *Figure 6—figure supplement 1*). One of the MDS genes with increased expression in *rcd1* encoded the sulfotransferase SOT12, an enzyme generating PAP. Accordingly, immunoblotting of total protein extracts with αSOT12 antibody demonstrated elevated SOT12 protein abundance in *rcd1* (*Figure 6B*). To address the functional interaction of RCD1 with PAP signaling, *rcd1-4* was crossed with *alx8* (also known as *sal1-8*). The resulting *rcd1 sal1* mutant was severely affected in development (*Figure 6C*). The effect of PAP signaling on the tolerance of PSII to chloroplastic ROS production was tested. The single *sal1* mutant was more tolerant to MV than Col-0, while under high MV concentration *rcd1 sal1* was even more MV-tolerant than *rcd1* (*Figure 6—figure supplement 3*). Together with transcriptomic similarities between *rcd1* and *sal1* mutants, these results further supported an overlap and/or synergy of PAP and RCD1 signaling pathways.

## RCD1 interacts with ANAC transcription factors in vivo

Expression of the MDS genes is regulated by the transcription factors ANAC013 and ANAC017 (*De Clercq et al., 2013*). The ANAC-responsive *cis*-element (*De Clercq et al., 2013*) was significantly enriched in promoter regions of *rcd1* mis-regulated genes (*Figure 6—figure supplement 1*). This suggested a functional connection between RCD1 and transcriptional regulation of the MDS genes by ANAC013/ANAC017. In an earlier study, ANAC013 was identified among many transcription factors interacting with RCD1 in the yeast two-hybrid system (*Jaspers et al., 2009*). This prompted us to investigate further the connection between RCD1 and ANAC013 and the in vivo relevance of this interaction.

Association of RCD1 with ANAC transcription factors in vivo was tested in two independent pull-down experiments. To identify interaction partners of ANAC013, an Arabidopsis line expressing ANAC013-GFP (*De Clercq et al., 2013*) was used. ANAC013-GFP was purified with αGFP beads, and associated proteins were identified by mass spectrometry in three replicates. RCD1 and its

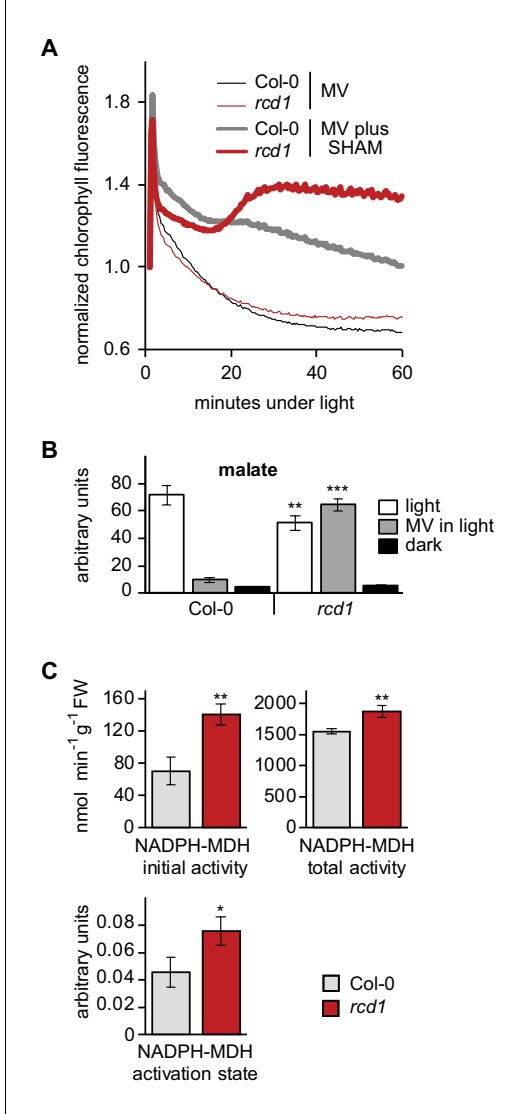

**Figure 5.** Altered electron transfer between the organelles in *rcd1*. (**A**) Leaf discs were pre-treated with 1 μM MV or MV plus 2 mM SHAM for 1 hr in the darkness. Then light was turned on (80 μmol m$^{-2}$ s$^{-1}$) and chlorophyll fluorescence under light (Fs) was recorded by Imaging PAM. Application of the two chemicals together caused Fs rise in *rcd1*, but not Col-0, suggesting increase in the reduction state of the chloroplast ETC in *rcd1*. For analysis of photochemical quenching see *Figure 5—figure supplement 1*. (**B**) Malate levels are significantly decreased in Col-0 but not in *rcd1* after MV treatment in light. Malate level was measured in extracts from Col-0 and *rcd1* seedlings that were pre-treated overnight with 50 μM MV or water control and collected either dark-adapted or after exposure to 4 hr of light. Mean ±SE are shown. Asterisks indicate values significantly different from those in the similarly treated wild type, ***P value < 0.001, **P value < 0.01, Student's t-test). For statistics, see *Figure 5—source data 1*. (**C**) NADPH-

*Figure 5 continued on next page*

closest homolog SRO1, as well as ANAC017, were identified as ANAC013 interacting proteins (see *Table 1* for a list of selected nuclear-localized interaction partners of ANAC013, and *Figure 7—source data 1* for the full list of identified proteins and mapped peptides). These data confirmed that ANAC013, RCD1 and ANAC017 are components of the same protein complex in vivo. In a reciprocal pull-down assay using transgenic Arabidopsis line expressing RCD1 tagged with triple Venus YFP under the control of *UBIQUITIN10* promoter, RCD1-3xVenus and interacting proteins were immunoprecipitated using αGFP (*Table 1*; *Figure 7—source data 2*). ANAC017 was found among RCD1 interactors.

To test whether RCD1 directly interacts with ANAC013/ANAC017 in vivo, the complex was reconstituted in the human embryonic kidney cell (HEK293T) heterologous expression system (details in *Figure 7—figure supplement 1*). Together with the results of in vivo pull-down assays, these experiments strongly supported the formation of a complex between RCD1 and ANAC013/ANAC017 transcription factors.

## Structural and functional consequences of RCD1-ANAC interaction

RCD1 interacts with many transcription factors belonging to different families (*Jaspers et al., 2009*; *Jaspers et al., 2010a*; *Vainonen et al., 2012*; *Bugge et al., 2018*) via its RST domain. The strikingly diverse set of RCD1 interacting partners may be partially explained by disordered flexible regions present in the transcription factors (*Kragelund et al., 2012*; *O'Shea et al., 2017*; *Bugge et al., 2018*). To address structural details of this interaction, the C-terminal domain of RCD1 (residues 468–589) including the RST domain (RST$_{RCD1}$; 510–568) was purified and labeled with $^{13}$C and $^{15}$N for NMR spectroscopic study (*Tossavainen et al., 2017*) (details in *Figure 7—figure supplement 2* and *Figure 7—source data 3*). ANAC013 was shown to interact with RCD1 in yeast two-hybrid assays (*Jaspers et al., 2009*; *O'Shea et al., 2017*). Thus, ANAC013$^{235-284}$ peptide was selected to address the specificity of the interaction of the RST domain with ANAC transcription factors using NMR (details in *Figure 7—figure supplement 3A,B*). Binding of RCD1$^{468-589}$ to ANAC013$^{235-284}$ caused profound changes in the HSQC spectrum of RCD1$^{468-589}$ (*Figure 7A* and *Figure 7—figure supplement 3C*). These data supported a strong and specific binary interaction between the RCD1

*Figure 5 continued*

MDH activity is increased in *rcd1*. To measure the activity of chloroplastic NADPH-MDH, plants were grown at 100–120 µmol m$^{-2}$ s$^{-1}$ at an 8 hr day photoperiod, leaves were collected in the middle of the day and freeze-dried. The extracts were prepared in the buffer supplemented with 250 µM thiol-reducing agent DTT, and initial activity was measured (top left). The samples were then incubated for 2 hr in the presence of additional 150 mM DTT, and total activity was measured (top right). The activation state of NADPH-MDH (bottom) is presented as the ratio of the initial and the total activity. Mean ±SE are shown. Asterisks indicate values significantly different from the wild type, **P value < 0.01, *P value < 0.05, Student's t-test. For statistics, see *Figure 5—source data 1*.
DOI: https://doi.org/10.7554/eLife.43284.022

The following source data and figure supplements are available for figure 5:

**Source data 1.** Source data and statistics.
DOI: https://doi.org/10.7554/eLife.43284.025
**Figure supplement 1.** Alterations in chloroplast electron transfer induced by MV and SHAM.
DOI: https://doi.org/10.7554/eLife.43284.023
**Figure supplement 2.** Distribution of malate in subcellular compartments of Col-0 and *rcd1*.
DOI: https://doi.org/10.7554/eLife.43284.024

RST domain and the ANAC013 transcription factor.

To evaluate the physiological significance of this interaction, stable *rcd1* complementation lines expressing an HA-tagged RCD1 variant lacking the C-terminus (amino acids 462–589) were generated. The *rcd1*: RCD1ΔRST-HA lines were characterized by increased accumulation of AOXs in comparison with the *rcd1*: RCD1-HA lines (*Figure 7B*). They also had *rcd1*-like tolerance of PSII to chloroplastic ROS (*Figure 7C*).

Physiological outcomes of the interaction between RCD1 and ANAC transcription factors were further tested by reverse genetics. ANAC017 regulates the expression of *ANAC013* in the mitochondrial retrograde signaling cascade (*Van Aken et al., 2016a*). Since ANAC017 precedes ANAC013 in the regulatory pathway and because no *anac013* knockout mutant is available, only the *rcd1-1 anac017* double mutant was generated. In the double mutant curly leaf habitus of *rcd1* was partially suppressed (*Figure 8A*). The *rcd1-1 anac017* mutant was more sensitive to chloroplastic ROS than the parental *rcd1* line (*Figure 8B*). The double mutant was characterized by lower abundance of AOX isoforms (*Figure 8C*), dramatically decreased expression of MDS genes (*Figure 8— figure supplement 1*) and lower AOX respiration

capacity (*Figure 8D*) compared to *rcd1*. Thus, gene expression, developmental, chloroplast- and mitochondria-related phenotypes of *rcd1* were partially mediated by ANAC017. These observations suggested that the in vivo interaction of RCD1 with ANAC transcription factors, mediated by the RCD1 C-terminal RST domain, is necessary for regulation of mitochondrial respiration and chloroplast ROS processing.

## Discussion

### RCD1 integrates chloroplast and mitochondrial signaling pathways

Plant chloroplasts and mitochondria work together to supply the cell with energy and metabolites. In these organelles, ROS are formed as by-products of the electron transfer chains (photosynthetic in chloroplasts and respiratory in mitochondria). ROS serve as versatile signaling molecules regulating many aspects of plant physiology such as development, stress signaling, systemic responses, and programmed cell death (PCD) (*Dietz et al., 2016*; *Noctor et al., 2018*; *Waszczak et al., 2018*). This communication network also affects gene expression in the nucleus where numerous signals are perceived and integrated. However, the molecular mechanisms of the coordinated action of the two energy organelles in response to environmental cues are only poorly understood. Evidence accumulated in this and earlier studies revealed the nuclear protein RCD1 as a regulator of energy organelle communication with the nuclear gene expression apparatus.

The *rcd1* mutant displays alterations in both chloroplasts and mitochondria (*Fujibe et al., 2004*; *Heiber et al., 2007*; *Jaspers et al., 2009*; *Broschė et al., 2014*; *Hiltscher et al., 2014*), and transcriptomic outcomes of RCD1 inactivation share similarities with those triggered by disrupted functions of both organelles (*Figure 6*). The results here suggest that RCD1 forms inhibitory complexes with components of mitochondrial retrograde signaling in vivo. Chloroplastic ROS appear to exhibit a direct influence on redox state and stability of RCD1 in the nucleus. These properties position RCD1 within a regulatory system encompassing mitochondrial complex III signaling through

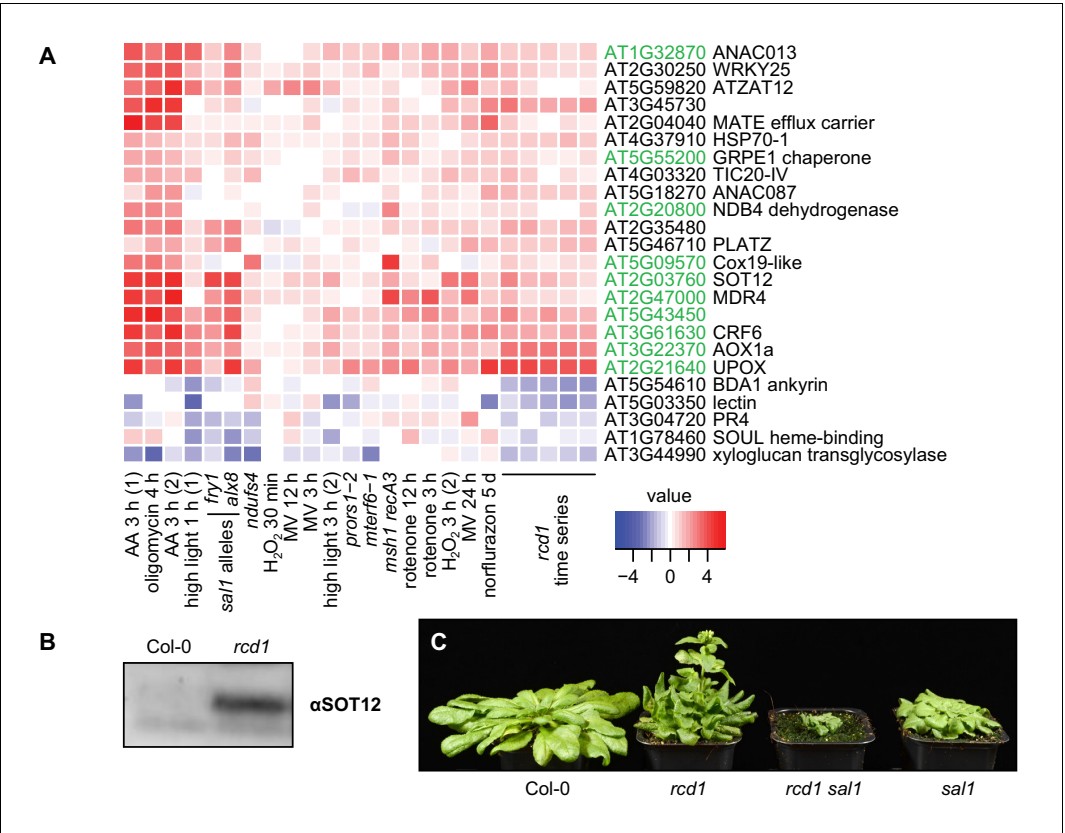

**Figure 6.** RCD1 is involved in mitochondrial dysfunction, chloroplast ROS and PAP signaling pathways. (**A**) Regulation of *rcd1* mis-expressed genes under perturbations of organellar functions in the selected subset of genes. A complete list of *rcd1*-misexpressed genes is presented in *Figure 6—figure supplement 1*. Similar transcriptomic changes are observed between the genes differentially regulated in *rcd1* and the genes affected by disturbed chloroplastic or mitochondrial functions. Mitochondrial dysfunction stimulon (MDS) genes regulated by ANAC013/ANAC017 transcription factors, are labeled green. (**B**) Sulfotransferase SOT12 encoded by an MDS gene accumulated in *rcd1* under standard growth conditions, as revealed by immunoblotting with the specific antibody. (**C**) Phenotype of the *rcd1 sal1* double mutant under standard growth conditions (12 hr photoperiod with white luminescent light of 220–250 µmol m$^{-2}$ s$^{-1}$).

DOI: https://doi.org/10.7554/eLife.43284.026

The following source data and figure supplements are available for figure 6:

**Source data 1.** Source data and statistics.
DOI: https://doi.org/10.7554/eLife.43284.030

**Figure supplement 1.** Clustering analysis of genes mis-regulated in *rcd1* (with cutoff of logFC <0.5) in published gene expression data sets acquired after perturbations of chloroplasts or mitochondria.
DOI: https://doi.org/10.7554/eLife.43284.027

**Figure supplement 2.** Induction of MDS genes in *rcd1* and *rcd1* complementation lines.
DOI: https://doi.org/10.7554/eLife.43284.028

**Figure supplement 3.** Tolerance of PSII to chloroplastic ROS in *sal1* mutants.
DOI: https://doi.org/10.7554/eLife.43284.029

ANAC013/ANAC017 transcription factors and chloroplastic signaling by $H_2O_2$. The existence of such an inter-organellar regulatory system, integrating mitochondrial ANAC013 and ANAC017-mediated signaling (*De Clercq et al., 2013*; *Ng et al., 2013b*) with the PAP-mediated chloroplastic signaling (*Estavillo et al., 2011*; *Chan et al., 2016*; *Crisp et al., 2018*) has been previously proposed on the basis of transcriptomic analyses (*Van Aken and Pogson, 2017*). However, the underlying molecular mechanisms remained unknown. Based on our results we propose that RCD1 may function at the intersection of mitochondrial and chloroplast signaling pathways and act as a nuclear integrator of both PAP and ANAC013 and ANAC017-mediated retrograde signals.

**Table 1.** Overview of the immunoprecipitation results.

Selected proteins identified in ANAC013-GFP and RCD1-3xVenus pull-down assays. Ratio vs. Col-0 and the P-value were obtained by Perseus statistical analysis from the three repeats for each genotype used. Bold text indicates baits. The peptide coverage for selected proteins as well as full lists of identified proteins are presented in *Figure 7—source datas 1* and *2*.

| Ratio vs. Col-0 | P-value | Unique peptides | Gene | Name | Stickiness |
|---|---|---|---|---|---|
| *ANAC013-GFP pull-down* | | | | | |
| 50966 | $7.09 \times 10^{-7}$ | 29 | AT1G32870 | **ANAC013** | |
| 22149 | $3.41 \times 10^{-8}$ | 25 | | **GFP** | |
| 10097 | $3.67 \times 10^{-6}$ | 37 | AT1G32230 | RCD1 | 1.00% |
| 110 | $1.67 \times 10^{-6}$ | 8 | AT2G35510 | SRO1 | 1.00% |
| 74 | $1.09 \times 10^{-9}$ | 4 | AT1G34190 | ANAC017 | 1.00% |
| *RCD1-3xVenus pull-down* | | | | | |
| 7593 | 0.000454 | 35 | AT1G32230 | **RCD1** | |
| 1292 | 0.006746 | 10 | | **YFP** | |
| 108 | $5.48 \times 10^{-8}$ | 2 | AT1G34190 | ANAC017 | 1.00% |

DOI: https://doi.org/10.7554/eLife.43284.031

RCD1 has been proposed to act as a transcriptional co-regulator because of its interaction with many transcription factors in yeast two-hybrid analyses (*Jaspers et al., 2009*). The in vivo interaction of RCD1 with ANAC013 and ANAC017 revealed in this study (*Table 1*, *Figures 7* and *8*) suggests that RCD1 modulates expression of the MDS, a set of ANAC013/ANAC017 activated nuclear genes mostly encoding mitochondrial components (*De Clercq et al., 2013*). *ANAC013* itself is an MDS gene, thus mitochondrial signaling through ANAC013/ANAC017 establishes a self-amplifying loop. Transcriptomic and physiological data support the role of RCD1 as a negative regulator of these transcription factors (*Figures 6–8*). Thus, RCD1 is likely involved in the negative regulation of the ANAC013/ANAC017 self-amplifying loop and in downregulating the expression of MDS genes after their induction.

Induction of genes in response to stress is commonly associated with rapid inactivation of a negative co-regulator. Accordingly, the RCD1 protein was sensitive to treatments triggering or mimicking chloroplastic ROS production. MV and $H_2O_2$ treatment of plants resulted in rapid oligomerization of RCD1 (*Figure 2*). Involvement of chloroplasts is indicated by the fact that MV treatment led to redox changes of RCD1-HA only in light (*Figure 2B* and *Figure 4—figure supplement 2B*). In addition, little change was observed with the mitochondrial complex III inhibitors AA or myx (*Figure 4—figure supplement 2A,B*). Together with the fact that MDS induction was not compromised in the *rcd1* mutant (*Figure 4—figure supplement 1* and *Figure 6—figure supplement 2*), this suggests that RCD1 may primarily function as a redox sensor of chloroplastic, rather than mitochondrial, ROS/redox signaling. In addition to fast redox changes, the overall level of RCD1 gradually decreased during prolonged (5 hr) stress treatments. This suggests several independent modes of RCD1 regulation at the protein level.

The complicated post-translational regulation of RCD1 is reminiscent of another prominent transcriptional co-regulator protein NONEXPRESSER OF PR GENES 1 (NPR1). NPR1 exists as a high molecular weight oligomer stabilized by intermolecular disulfide bonds between conserved cysteine residues. Accumulation of salicylic acid and cellular redox changes lead to the reduction of cysteines and release of NPR1 monomers that translocate to the nucleus and activate expression of defense genes (*Kinkema et al., 2000*; *Mou et al., 2003*; *Withers and Dong, 2016*). Similar to NPR1, RCD1 has a bipartite nuclear localization signal and, in addition, a putative nuclear export signal between the WWE and PARP-like domains. Like NPR1, RCD1 has several conserved cysteine residues. Interestingly, mutation of seven interdomain cysteines in RCD1 largely eliminated the fast in vivo effect of chloroplastic ROS on redox state and stability of RCD1; however, it did not significantly alter the plant response to MV (*Figure 2* and *Figure 2—figure supplement 1C,D*). This suggests that redox-dependent oligomerization of RCD1 may serve to fine-tune its activity.

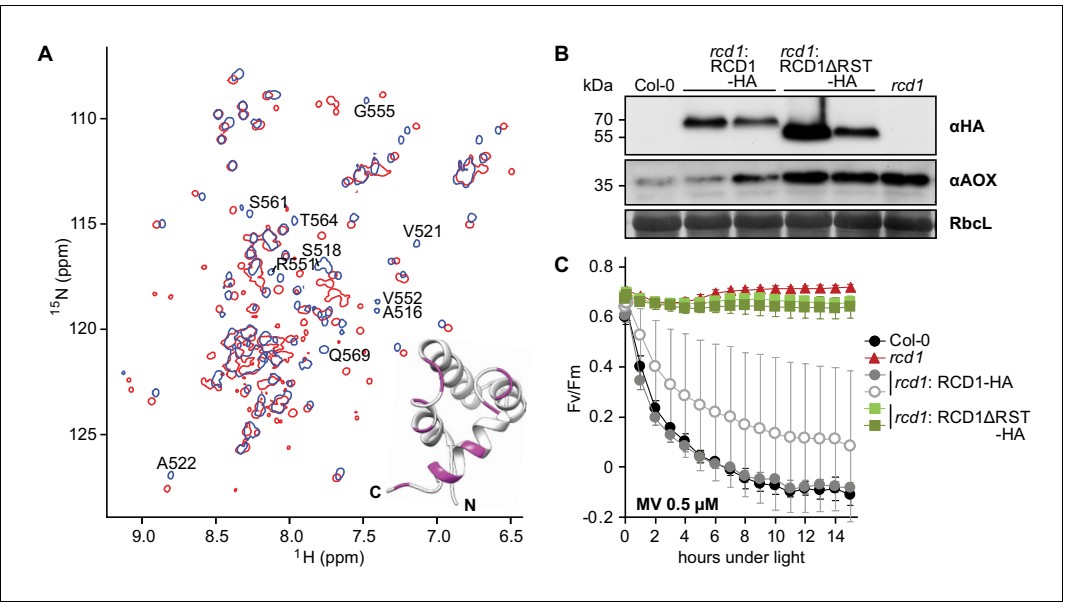

**Figure 7.** RST domain of RCD1 binds to ANAC transcription factors and is necessary for RCD1 function in vivo. Source data and statistics are presented in *Figure 7—source data 4*. (**A**) Biochemical interaction of ANAC013 with the RST domain of RCD1 in vitro. Superimposed $^1H$, $^{15}N$ HSQC spectra of the C-terminal domain of RCD1 acquired in absence (blue) and presence (red) of approximately two-fold excess of the ANAC013$^{235-284}$ peptide. Interaction of RCD1$^{468-589}$ with ANAC013$^{235-284}$ caused peptide-induced chemical shift changes in the $^1H$, $^{15}N$ correlation spectrum of RCD1, which were mapped on the structure of the RST domain (inset). Inset: RST$_{RCD1}$ structure with highlighted residues demonstrating the largest chemical shift perturbations ($\Delta\delta \geq 0.10$ ppm) between the free and bound forms (details in *Figure 7—figure supplement 3C*), which probably corresponds to ANAC013-interaction site. (**B**) Stable expression in *rcd1* of the HA-tagged RCD1 variant lacking its C-terminus under the control of the native *RCD1* promoter does not complement *rcd1* phenotypes. In the independent complementation lines RCD1ΔRST-HA was expressed at the levels comparable to those in the RCD1-HA lines (upper panel). However, in *rcd1*: RCD1ΔRST-HA lines abundance of AOXs (middle panel) was similar to that in *rcd1*. (**C**) Tolerance of PSII to chloroplastic ROS was similar in the *rcd1*: RCD1ΔRST-HA lines and *rcd1*. For each PSII inhibition experiment, leaf discs from at least four individual rosettes were used. The experiment was performed three times with similar results. Mean ±SD are shown.

DOI: https://doi.org/10.7554/eLife.43284.032

The following source data and figure supplements are available for figure 7:

**Source data 1.** In vivo interaction partners of ANAC013.
DOI: https://doi.org/10.7554/eLife.43284.036
**Source data 2.** In vivo interaction partners of RCD1.
DOI: https://doi.org/10.7554/eLife.43284.037
**Source data 3.** NMR constraints and structural statistics for the ensemble of the 15 lowest-energy structures of RCD1 RST.
DOI: https://doi.org/10.7554/eLife.43284.038
**Source data 4.** Source data and statistics.
DOI: https://doi.org/10.7554/eLife.43284.039
**Figure supplement 1.** Biochemical interaction of RCD1 with ANAC013/ANAC017 transcription factors in human embryonic kidney (HEK293) cells.
DOI: https://doi.org/10.7554/eLife.43284.033
**Figure supplement 2.** Structure of the RST domain of RCD1.
DOI: https://doi.org/10.7554/eLife.43284.034
**Figure supplement 3.** Analysis of interaction of the ANAC013-derived peptides with the RST domain of RCD1.
DOI: https://doi.org/10.7554/eLife.43284.035

## MDS genes are involved in interactions between the organelles

How the RCD1-dependent induction of MDS genes contributes to the energetic and signaling landscape of the plant cell remains to be investigated. Our results suggest that one component of this

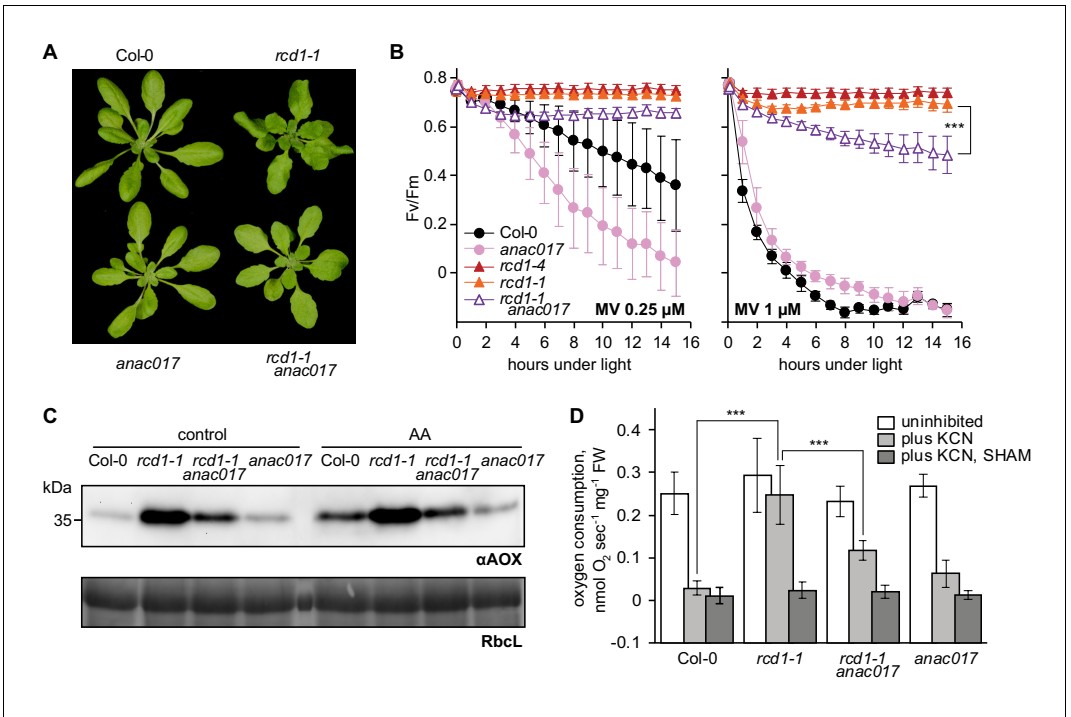

**Figure 8.** Developmental, chloroplast- and mitochondria-related phenotypes of *rcd1* are partially mediated by ANAC017. Source data and statistics are presented in *Figure 8—source data 1*. (**A**) Introducing *anac017* mutation in the *rcd1* background partially suppressed the curly leaf phenotype of *rcd1*. (**B**) The *anac017* mutation partially suppressed tolerance of *rcd1* to chloroplastic ROS. PSII inhibition by ROS was measured in *rcd1 anac017* double mutant by using 0.25 µM or 1 µM MV (left and right panel, accordingly). For each experiment, leaf discs from at least four individual rosettes were used. The experiment was performed three times with similar results. Mean ±SD are shown. Asterisks denote selected values that are significantly different (P value < 0.001, two-way ANOVA with Bonferroni post hoc correction). (**C**) The *anac017* mutation partially suppressed mitochondrial phenotypes of *rcd1*. Total AOX protein levels were lowered in *rcd1 anac017* double mutant as compared to *rcd1* both after the overnight treatment with 2.5 µM AA and in the untreated control. (**D**) Oxygen uptake by *rcd1 anac017* seedlings was measured in the darkness in presence of mitochondrial respiration inhibitors as described in *Figure 4C*. The *rcd1 anac017* mutant demonstrated lower KCN-insensitive AOX respiration capacity than *rcd1*. Each measurement was performed on 10–15 pooled seedlings and repeated at least three times. Mean ±SD are shown. Asterisks denote selected values that are significantly different (P value < 0.001, one-way ANOVA with Bonferroni post hoc correction).

DOI: https://doi.org/10.7554/eLife.43284.040

The following source data and figure supplement are available for figure 8:

**Source data 1.** Source data and statistics.
DOI: https://doi.org/10.7554/eLife.43284.042

**Figure supplement 1.** Induction of MDS genes in *anac017* and *rcd1 anac017* mutants.
DOI: https://doi.org/10.7554/eLife.43284.041

adaptation is the activity of mitochondrial alternative oxidases, which are part of the MDS regulon. Consequently, AOX proteins accumulate at higher amounts in *rcd1* (*Figure 4*). Pretreatment of wild type plants with complex III inhibitors AA or myx led to elevated AOX abundance coinciding with increased tolerance to chloroplastic ROS. Moreover, the AOX inhibitor SHAM made plants more sensitive to MV, indicating the direct involvement of AOX activity in the chloroplastic ROS processing. It thus appears that AOXs in the mitochondria form an electron sink that indirectly contributes to the oxidization of the electron acceptor side of PSI. In the *rcd1* mutant, this mechanism may be continuously active. The described inter-organellar electron transfer may decrease production of ROS by PSI (asterisk in *Figure 9*). Furthermore, chloroplastic ROS are considered the main electron sink for oxidation of chloroplast thiol enzymes (*Ojeda et al., 2018*; *Vaseghi et al., 2018*; *Yoshida et al., 2018*). Thus, the redox status of these enzymes could depend on the proposed inter-

organellar pathway. This is in line with higher reduction of the chloroplast enzymes 2-CP and NADPH-MDH observed in *rcd1* (*Figure 1C* and *Figure 5C*).

The malate shuttle was recently shown to mediate a chloroplast-to-mitochondria electron transfer pathway that caused ROS production by complex III and evoked mitochondrial retrograde signaling (*Wu et al., 2015*; *Zhao et al., 2018*). Altered levels of malate and increased activity of NADPH-dependent malate dehydrogenase in *rcd1* (*Figure 5*) suggest that in this mutant the malate shuttle could act as an inter-organellar electron carrier.

Another MDS gene with more abundant mRNA levels in the *rcd1* mutant encodes sulfotransferase SOT12, an enzyme involved in PAP metabolism (*Klein and Papenbrock, 2004*). Accordingly, SOT12 protein level was significantly increased in the *rcd1* mutant (*Figure 6B*). Accumulation of SOT12 and similarities between transcript profiles of RCD1- and PAP-regulated genes suggest that PAP signaling is likely to be constitutively active in the *rcd1* mutant. Unbalancing this signaling by elimination of SAL1 leads to severe developmental defects, as evidenced by the stunted phenotype of the *rcd1 sal1* double mutant. Thus, the RCD1 and the PAP signaling pathways appear to be overlapping and somewhat complementary, but the exact molecular mechanisms remain to be explored.

## RCD1 regulates stress responses and cell fate

The MDS genes represent only a fraction of genes showing differential regulation in *rcd1* (*Figure 6—figure supplement 1*). This likely reflects the fact that RCD1 interacts with many other protein partners in addition to ANACs. The C-terminal RST domain of RCD1 was shown to interact with transcription factors belonging to DREB, PIF, ANAC, Rap2.4 and other families (*Jaspers et al., 2009*; *Vainonen et al., 2012*; *Hiltscher et al., 2014*; *Bugge et al., 2018*). Analyses of various transcription factors interacting with RCD1 revealed little structural similarity between their RCD1-interacting sequences (*O'Shea et al., 2017*). The flexible structure of the C-terminal domain of RCD1 probably determines the specificity and ability of RCD1 to interact with those different transcription factors. This makes RCD1 a hub in the crosstalk of organellar signaling with hormonal, photoreceptor, immune and other pathways and a likely mechanism by which these pathways are integrated and co-regulated.

The changing environment requires plants to readjust continuously their energy metabolism and ROS processing. On the one hand, this happens because of abiotic stress factors such as changing light intensity or temperature. For example, a sunlight fleck on a shade-adapted leaf can instantly alter excitation pressure on photosystems by two orders of magnitude (*Allahverdiyeva et al., 2015*). On the other hand, chloroplasts and mitochondria are implicated in plant immune reactions to pathogens, contributing to decisive checkpoints including PCD (*Shapiguzov et al., 2012*; *Petrov et al., 2015*; *Wu et al., 2015*; *Van Aken and Pogson, 2017*; *Zhao et al., 2018*). In both scenarios, perturbations of organellar ETCs may be associated with increased production of ROS. However, the physiological outcomes of the two situations can be opposite: acclimation in one case and cell death in the other. The existence of molecular mechanisms that unambiguously differentiate one type of response from the other has been previously suggested (*Trotta et al., 2014*; *Sowden et al., 2018*; *Van Aken and Pogson, 2017*). The ANAC017 transcription factor and

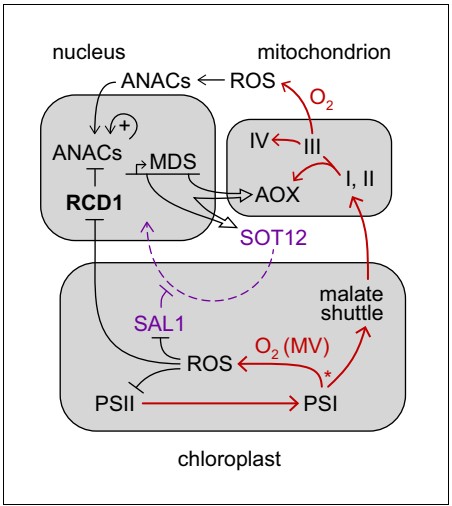

**Figure 9.** Hypothetical role of RCD1 in organelle signaling and energy metabolism. RCD1 is the direct suppressor of ANAC transcription factors that is itself subject to redox regulation. Chloroplastic ROS likely affect RCD1 protein redox state and abundance. Inactivation of RCD1 leads to induction of ANAC-controlled MDS regulon. Expression of MDS genes is possibly feedback-regulated *via* the PAP retrograde signaling (purple). Resulting activation of mitochondrial AOXs and other MDS components is likely to affect electron flows (red) and ROS signaling in mitochondria and in chloroplasts. Putative competition of AOX-directed electron transfer with the formation of ROS at PSI is labeled with an asterisk.

DOI: https://doi.org/10.7554/eLife.43284.043

MDS genes, as well as PAP signaling, were proposed as organelle-related components counteracting PCD during abiotic stress (*Van Aken and Pogson, 2017*). This suggests that RCD1 is involved in the regulation of the cell fate checkpoint. Accordingly, the *rcd1* mutant is resistant to a number of abiotic stress treatments (*Ahlfors et al., 2004*; *Fujibe et al., 2004*; *Jaspers et al., 2009*).

Interestingly, in contrast to its resistance to abiotic stress, *rcd1* is more sensitive to treatments related to biotic stress. The *rcd1* mutant was originally identified in a forward genetic screen for sensitivity to ozone (*Overmyer et al., 2000*). Ozone decomposes in the plant cell wall to ROS mimicking formation of ROS by respiratory burst oxidases (RBOHs) in the course of plant immune reactions (*Joo et al., 2005*; *Vainonen and Kangasjärvi, 2015*). The opposing roles of RCD1 in the cell fate may be related to its interaction with diverse transcription factor partners and/or different regulation of its stability and abundance. For example, transcriptomic analyses showed that under standard growth conditions, a cluster of genes associated with defense against pathogens had decreased expression in *rcd1* (*Brosché et al., 2014*), and no ANAC013/ANAC017 *cis*-element motif is associated with these genes (*Figure 6—figure supplement 1*). In agreement with its role in biotic stress, RCD1 is a target for a fungal effector protein that prevents the activation of plant immunity (*Wirthmueller et al., 2018*).

Another possible factor determining varying roles of RCD1 in the cell fate is differential regulation of RCD1 protein function by ROS/redox signals emitted by different subcellular compartments. The sensitivity of RCD1 to chloroplastic ROS (*Figure 2*) can be interpreted as negative regulation of the pro-PCD component. We hypothesize that this inactivation can occur in environmental situations that require physiological adaptation rather than PCD. For example, an abrupt increase in light intensity can cause excessive electron flow in photosynthetic ETC and overproduction of reducing power. The resulting deficiency of PSI electron acceptors can lead to changes in chloroplastic ROS production, which *via* retrograde signaling might influence RCD1 stability and/or redox status, inhibiting its activity and thus affecting adjustments in nuclear gene expression (*Figure 9*). Among other processes, RCD1-mediated suppression of ANAC013/ANAC017 transcription factors is released, allowing the induction of the MDS regulon. The consequent expression of AOXs together with increased chloroplast-to-mitochondrial electron transfer is likely to provide electron sink for photosynthesis, which could suppress chloroplast ROS production and contribute to the plant's survival under a changing environment (*Figure 9*).

## Materials and methods

**Key resources table**

| Reagent type (species) or resource | Designation | Source or reference | Identifiers | Additional information |
|---|---|---|---|---|
| Genetic reagent (*Arabidopsis thaliana*, Col-0) | *rcd1-4* | NASC stock center | GK-229D11 | homozygous mutant plant line |
| Genetic reagent (*Arabidopsis thaliana*, Col-0) | *rcd1-1* | PMID: 11041881 | | homozygous mutant plant line |
| Genetic reagent (*Arabidopsis thaliana*, Col-0) | *aox1a* | PMID: 16299171 | | homozygous mutant plant line |
| Genetic reagent (*Arabidopsis thaliana*, Col-0) | *AOX1a*-OE | PMID: 16299171 | | homozygous mutant plant line |
| Genetic reagent (*Arabidopsis thaliana*, Col-0) | *ptox* | PMID: 7920709 | | homozygous mutant plant line |
| Genetic reagent (*Arabidopsis thaliana*, Col-0) | *anac017* | NASC stock center | SALK_022174 | homozygous mutant plant line |
| Genetic reagent (*Arabidopsis thaliana*, Col-0) | *sal1-8* | PMID: 19170934 | | homozygous mutant plant line |
| Genetic reagent (*Arabidopsis thaliana*, Col-0) | *rcd1 aox1a* | PMID: 24550736 | | homozygous mutant plant line |

*Continued on next page*

*Continued*

| Reagent type (species) or resource | Designation | Source or reference | Identifiers | Additional information |
|---|---|---|---|---|
| Genetic reagent (*Arabidopsis thaliana*, Col-0) | *rcd1-1 anac017* | this paper | | homozygous mutant plant line |
| Genetic reagent (*Arabidopsis thaliana*, Col-0) | *rcd1-4 sal1-8* | this paper | | homozygous mutant plant line |
| Genetic reagent (*Arabidopsis thaliana*, Col-0) | *rcd1-4*: RCD1-HA | this paper | | set of complementation plant lines |
| Genetic reagent (*Arabidopsis thaliana*, Col-0) | *rcd1-4:* RCD1-3xVenus | this paper | | set of complementation plant lines |
| Genetic reagent (*Arabidopsis thaliana*, Col-0) | *rcd1-4*: RCD1Δ7Cys-HA | this paper | | set of complementation plant lines |
| Genetic reagent (*Arabidopsis thaliana*, Col-0) | *rcd1-4*: RCD1ΔRST-HA | this paper | | set of complementation plant lines |
| Genetic reagent (*Arabidopsis thaliana*, Col-0) | ANAC013-GFP | PMID: 24045019 | | transgenic plant line |
| Genetic reagent (*Arabidopsis thaliana*, *gl*1) | *pgr5* | PMID: 12176323 | | homozygous mutant plant line |
| Cell line (*Homo sapiens*) | HEK293T | ATCC | ATCC CRL-3216 | human embryonic kidney cell line |
| Gene (*Homo sapiens*) | HA-RCD1 | this paper | | construct for expression in HEK293T cells |
| Gene (*Homo sapiens*) | ANAC013-myc | this paper | | construct for expression in HEK293T cells |
| Gene (*Homo sapiens*) | ANAC017-myc | this paper | | construct for expression in HEK293T cells |
| Antibody | αHA | Roche | Roche 1 867 423 001 | 1: 2 000 for immunoblotting |
| Antibody | αGFP | Milteny Biotech | | |
| Antibody | αRCD1 | this paper | | 1: 500 for immunoblotting |
| Antibody | αSOT12 | Dr. Saijaliisa Kangasjärvi | Agrisera AS16 3943 | 1: 500 for immunoblotting |
| Peptide, recombinant protein | ANAC013 peptides | Genecust | | Synthetic peptides |

## Plants and mutants

*Arabidopsis thaliana* adult plants were grown on soil (peat: vermiculite = 1:1) in white luminescent light (220–250 µmol m$^{-2}$ s$^{-1}$) at a 12 hr photoperiod. Seedlings were grown for 14 days on 1 x MS basal medium (Sigma-Aldrich) with 0.5% Phytagel (Sigma-Aldrich) without added sucrose in white luminescent light (150–180 µmol m$^{-2}$ s$^{-1}$) at a 12 hr photoperiod. Arabidopsis *rcd1-4* mutant (GK-229D11), *rcd1-1* (*Overmyer et al., 2000*), *aox1a* (SAIL_030_D08), *AOX1a*-OE (*Umbach et al., 2005*), *ptox* (*Wetzel et al., 1994*), *anac017* (SALK_022174), and *sal1-8* (*Wilson et al., 2009*) mutants are of Col-0 background; *pgr5* mutant is of *gl1* background (*Munekage et al., 2002*). ANAC013-GFP line is described in *De Clercq et al. (2013)*, RCD1-HA line labeled 'a' in *Figure 1—figure supplement 1* is described in *Jaspers et al. (2009)*, *rcd1 aox1a* double mutant – in *Brosché et al. (2014)*. RCD1-3xVenus, RCD1Δ7Cys-HA, RCD1ΔRST-HA lines are described in *Cloning*.

## Cloning

The *rcd1* complementation line expressing RCD1 tagged with triple HA epitope on the C-terminus was described previously (*Jaspers et al., 2009*). In this line the genomic sequence of RCD1 was expressed under the control of the *RCD1* native promotor (3505 bp upstream the start codon). The RCD1Δ7Cys-HA construct was generated in the same way as RCD1-HA. The cysteine residues were mutated to alanines by sequential PCR-based mutagenesis of the genomic sequence of *RCD1* in the pDONR/Zeo vector followed by end-joining with In-Fusion (Clontech). The RCD1ΔRST-HA variant was generated in the same vector by removal with a PCR reaction of the region corresponding to

amino acid residues 462–589. The resulting construct was transferred to the pGWB13 binary vector by a Gateway reaction. To generate the RCD1-3xVenus construct, RCD1 cDNA was fused to the *UBIQUITIN10* promoter region and to the C-terminal triple Venus YFP tag in a MultiSite Gateway reaction as described in *Siligato et al. (2016)*. The vectors were introduced in the *rcd1-4* mutant by floral dipping. Homozygous single insertion Arabidopsis lines were obtained. They were defined as the lines demonstrating 1:3 segregation of marker antibiotic resistance in T2 generation and 100% resistance to the marker antibiotic in T3 generation.

For HEK293T cell experiments codon-optimized N-terminal 3xHA-fusion of RCD1 and C-terminal 3xmyc-fusion of ANAC013 were cloned into pcDNA3.1(+). Full-length ANAC017 was cloned into pcDNA3.1(-) in the Xho I/Hind III sites, the double myc tag was introduced in the reverse primer sequence. The primer sequences used for the study are presented in *Supplementary file 1*.

## Generation of the αRCD1 antibody

αRCD1 specific antibody was raised in rabbit using denatured RCD1-6His protein as the antigen for immunization (Storkbio, Estonia). The final serum was purified using denatured RCD1-6His immobilized on nitrocellulose membrane, aliquoted and stored at −80°C. For immunoblotting, 200 μg of total protein were loaded per well, the antibody was used in dilution 1: 500.

## Inhibitor treatments

For PSII inhibition studies, leaf discs were let floating on Milli-Q water solution supplemented with 0.05% Tween 20 (Sigma-Aldrich). Final concentration of AA and myx was 2.5 μM each, of SHAM – 2 mM. For transcriptomic experiments, plant rosettes were sprayed with water solution of 50 μM AA complemented with 0.01% Silwet Gold (Nordisk Alkali). Stock solutions of these chemicals were prepared in DMSO, equal volumes of DMSO were added to control samples. Pre-treatment with chemicals was carried out in the darkness, overnight for MV, AA and myx, 1 hr for SHAM. After spraying plants with 50 μM AA they were incubated in growth light for 3 hr. For chemical treatment in seedlings grown on MS plates, 5 mL of Milli-Q water with or without 50 μM MV were poured in 9 cm plates at the end of the light period. The seedlings were kept in the darkness overnight, and light treatment was performed on the following morning. For $H_2O_2$ treatment, the seedlings were incubated in 5 mL of Milli-Q water with or without 100 mM $H_2O_2$ in light.

## DAB staining

Plant rosettes were stained with 3,3′-diaminobenzidine (DAB) essentially as described in *Daudi et al. (2012)*. After vacuum infiltration of DAB-staining solution in the darkness, rosettes were exposed to light (180 μmol m$^{-2}$ s$^{-1}$) for 20 min to induce production of chloroplastic ROS and then immediately transferred to the bleaching solution.

## Spectroscopic measurements of photosynthesis

Chlorophyll fluorescence was measured by MAXI Imaging PAM (Walz, Germany). PSII inhibition protocol consisted of repetitive 1 hr periods of blue actinic light (450 nm, 80 μmol m$^{-2}$ s$^{-1}$) each followed by a 20 min dark adaptation, then Fo and Fm measurement. PSII photochemical yield was calculated as Fv/Fm = (Fm-Fo)/Fm (*Figure 1—figure supplement 2*). To plot raw chlorophyll fluorescence kinetics under light (Fs) against time, the reads were normalized to dark-adapted Fo. For the measurements of photochemical quenching, Fm′ was measured with saturating pulses triggered against the background of actinic light (450 nm, 80 μmol m$^{-2}$ s$^{-1}$), and the following formulae were used: qP = (Fm′ - Fs)/(Fm′-Fo′), where Fo′ ≈ Fo / (((Fm − Fo)/Fm) + (Fo/Fm′)) (*Oxborough and Baker, 1997*). The assays were performed in 96-well plates. In each assay, leaf discs from at least four individual plants were analyzed. Each assay was reproduced at least three times.

PSI (P700) oxidation was measured by DUAL-PAM-100 (Walz, Germany) as described (*Tiwari et al., 2016*). Leaves were pre-treated in 1 μM MV for 4 hr, then shifted to light (160 μmol m$^{-2}$ s$^{-1}$) for indicated time. Oxidation of P700 was induced by PSI-specific far red light (FR, 720 nm). To determine fully oxidized P700 (Pm), a saturating pulse of actinic light was applied under continuous background of FR, followed by switching off both the actinic and FR light. The kinetics of P700$^{+}$ reduction by intersystem electron transfer pool and re-oxidation by FR was determined by using a multiple turnover saturating flash of PSII light (635 nm) in the background of continuous FR.

## Isolation, separation and detection of proteins and protein complexes

Thylakoids were isolated as described in *Järvi et al. (2016)*. Chlorophyll content was determined according to *Porra et al. (1989)* and protein content according to *Lowry et al. (1951)*. For immunoblotting of total plant extracts, the plant material was frozen immediately after treatments in liquid nitrogen and ground. Total proteins were extracted in SDS extraction buffer [50 mM Tris-HCl (pH 7.8), 2% SDS, 1 x protease inhibitor cocktail (Sigma-Aldrich), 2 mg/mL NaF] for 20 min at 37°C and centrifuged at 18 000 x *g* for 10 min. Supernatants were normalized for protein concentration and resolved by SDS-PAGE. For separation of proteins, SDS-PAGE (10–12% polyacrylamide) was used (*Laemmli, 1970*). For thylakoid proteins, the gel was complemented with 6 M urea. To separate thylakoid membrane protein complexes, isolated thylakoids were solubilized with *n*-dodecyl β-D-maltoside (Sigma-Aldrich) and separated in BN-PAGE (5–12.5% polyacrylamide) as described by *Järvi et al. (2016)*. After electrophoresis, proteins were electroblotted to PVDF membrane and immunoblotted with specific antibodies. αSOT12 antibodies have Agrisera reference number AS16 3943. For quantification of immunoblotting signal, ImageJ software was used (https://imagej.nih.gov/ij/).

## Analysis of protein thiol redox state by mobility shift assays

Thiol redox state of 2-CPs in detached Col-0 and *rcd1* leaves adapted to darkness or light (3 hr of 160 µmol m$^{-2}$ s$^{-1}$), was determined by alkylating free thiols in TCA-precipitated proteins with 50 mM N-ethylmaleimide in the buffer containing 8 M urea, 100 mM Tris-HCl (pH 7.5), 1 mM EDTA, 2% SDS, and 1/10 of protease inhibitor cocktail (Thermo Scientific), reducing in vivo disulfides with 100 mM DTT and then alkylating the newly reduced thiols with 10 mM methoxypolyethylene glycol maleimide of molecular weight 5 kDa (Sigma-Aldrich), as described in *Nikkanen et al. (2016)*. Proteins were then separated by SDS-PAGE and immunoblotted with a 2-CP-specific antibody.

## Non-aqueous fractionation (NAF)

Leaves of Arabidopsis plants were harvested in the middle of the light period and snap-frozen in liquid nitrogen. Four grams of fresh weight of frozen plant material was ground to a fine powder using a mixer mill (Retsch), transferred to Falcon tubes and freeze-dried at 0.02 bar for 5 days in a lyophilizer, which had been pre-cooled to −40°C. The NAF-fractionation procedure was performed as described in *Krueger et al. (2011)*, *Arrivault et al. (2014)*, and *Krueger et al. (2014)*, except that the gradient volume, composed of the solvents tetrachloroethylene ($C_2Cl_4$)/heptane ($C_7H_{16}$), was reduced from 30 mL to 25 mL but with the same linear density. Leaf powder was resuspended in 20 mL $C_2Cl_4$/$C_7H_{16}$ mixture 66:34 (v/v; density ρ = 1.3 g cm$^{-3}$), and sonicated for 2 min, with 6 × 10 cycles at 65% power. The sonicated suspension was filtered through a nylon net (20 µm pore size). The net was washed with 30 mL of heptane. The suspension was centrifuged for 10 min at 3 200 x *g* at 4°C and the pellet was resuspended in 5 mL $C_2Cl_4$/$C_7H_{16}$ mixture 66:34. The gradient was formed in 38 mL polyallomer centrifugation tube using a peristaltic gradient pump (BioRad) generating a linear gradient from 70% solvent A ($C_2Cl_4$/$C_7H_{16}$ mixture 66:34) to 100% solvent B (100% $C_2Cl_4$) with a flow rate of 1.15 mL min$^{-1}$, resulting in a density gradient from 1.43 g cm$^{-3}$ to 1.62 g cm$^{-3}$. Five mL suspension containing the sample was loaded on top of the gradient and centrifuged for 55 min at 5 000 x *g* at 4°C using a swing-out rotor with acceleration and deceleration of 3:3 (brakes off). Each of the compartment-enriched fractions (F1 to F8) were transferred carefully from the top of the gradient into a 50 mL Falcon tube, filled up with heptane to a volume of 20 mL and centrifuged at 3 200 x *g* for 10 min. The pellet was resuspended in 6 mL of heptane and subsequently divided into 6 aliquots of equal volume (950 µL). The pellets had been dried in a vacuum concentrator without heating and stored at −80°C until further use. Subcellular compartmentation of markers or the metabolites of our interest was calculated by BestFit method as described in *Krueger et al. (2011)* and *Krueger et al. (2014)*. Percentage values (% of the total found in all fractions) of markers and metabolites have been used to make the linear regressions for subcellular compartments using BestFit.

## Marker measurements for non-aqueous fractionation

Before enzyme and metabolite measurements, dried pellets were homogenized in the corresponding extraction buffer by the addition of one steel ball (2 mm diameter) to each sample and shaking

at 25 Hz for 1 min in a mixer mill. Enzyme extracts were prepared as described in *Gibon et al. (2004)* with some modifications. The extraction buffer contained 50 mM HEPES-KOH (pH 7.5), 10 mM $MgCl_2$, 1 mM EDTA, 1 mM EGTA, 1 mM benzamidine, 1 mM ε-aminocaproic acid, 0.25% (w/v) BSA, 20 μM leupeptin, 0.5 mM DTT, 1 mM phenylmethylsulfonyl fluoride (PMSF), 1% (v/v) Triton X-100, 20% glycerol. The extract was centrifuged (14 000 rpm at 4°C for 10 min) and the supernatant was used directly for the enzymatic assays. The activities of adenosine diphosphate glucose pyrophosphorylase (AGPase) and phosphoenolpyruvate carboxylase (PEPC) were determined as described in *Gibon et al. (2004)* but without using the robot-based platform. Chlorophyll was extracted twice with 80% (v/v) and once with 50% (v/v) hot ethanol/10 mM HEPES (pH 7.0) followed by 30 min incubation at 80°C and determined as described in *Cross et al. (2006)*. Nitrate was measured by the enzymatic reaction as described in *Cross et al. (2006)*.

## Incubation of Arabidopsis leaf discs with [U-$^{14}$C] glucose

For the light experiment, leaf discs were incubated in light in 5 mL 10 mM MES-KOH (pH 6.5), containing 1.85 MBq/mmol [U-$^{14}$C] glucose (Hartmann Analytic) in a final concentration of 2 mM. In the dark experiment, leaf discs were incubated under green light for 150 min. Leaf discs were placed in a sieve, washed several times in double-distilled water, frozen in liquid nitrogen, and stored at −80°C until further analysis. All incubations were performed in sealed flasks under green light and shaken at 100 rpm. The evolved $^{14}CO_2$ was collected in 0.5 mL of 10% (w/v) KOH.

## Fractionation of $^{14}$C-labeled tissue extracts and measurement of metabolic fluxes

Extraction and fractionation were performed according to *Obata et al. (2017)*. Frozen leaf discs were extracted with 80% (v/v) ethanol at 80°C (1 mL per sample) and re-extracted in two subsequent steps with 50% (v/v) ethanol (1 mL per sample for each step), and the combined supernatants were dried under an air stream at 35°C and resuspended in 1 mL of water (*Fernie et al., 2001*). The soluble fraction was subsequently separated into neutral, anionic, and basic fractions by ion-exchange chromatography; the neutral fraction (2.5 mL) was freeze-dried, resuspended in 100 μL of water, and further analyzed by enzymatic digestion followed by a second ion-exchange chromatography step (*Carrari et al., 2006*). To measure phosphate esters, samples (250 μL) of the soluble fraction were incubated in 50 μL of 10 mM MES-KOH (pH 6.0), with or without 1 unit of potato acid phosphatase (grade II; Boehringer Mannheim) for 3 hr at 37°C, boiled for 2 min, and analyzed by ion-exchange chromatography (*Fernie et al., 2001*). The insoluble material left after ethanol extraction was homogenized, resuspended in 1 mL of water, and counted for starch (*Fernie et al., 2001*). Fluxes were calculated as described following the assumptions detailed by *Geigenberger et al. (1997)* and *Geigenberger et al. (2000)*. Unfortunately, the discontinued commercial availability of the required positionally radiolabeled glucoses prevented us from analyzing fermentative fluxes more directly.

## Preparation of crude mitochondria

Crude mitochondria were isolated from Arabidopsis rosette leaves as described in *Keech et al. (2005)*.

## Measurements of AOX capacity in vivo

Seedling respiration and AOX capacity were assessed by measuring $O_2$ consumption in the darkness using a Clark electrode as described in *Schwarzländer et al. (2009)*.

## Metabolite extraction

Primary metabolites were analyzed with GC-MS according to *Roessner et al. (2000)*. GC-MS analysis was executed from the plant extracts of eight biological replicates (pooled samples). Plant material was homogenized in a Qiagen Tissuelyser II bead mill (Qiagen, Germany) with 1–1.5 mm Retsch glass beads. Soluble metabolites were extracted from plant material in two steps, first with 1 mL of 100% methanol (Merck) and second with 1 mL of 80% (v/v) aqueous methanol. During the first extraction step, 5 μL of internal standard solution (0.2 mg mL$^{-1}$ of benzoic-d$_5$ acid, 0.1 mg mL$^{-1}$ of glycerol-d$_8$, 0.2 mg mL$^{-1}$ of 4-methylumbelliferone in methanol) was added to each sample. During both extraction steps, the samples were vortexed for 30 min and centrifuged for 5 min at 13 000

rpm (13 500 × *g*) at 4°C. The supernatants were then combined for metabolite analysis. The extracts (2 mL) were dried in a vacuum concentrator (MiVac Duo, Genevac Ltd, Ipswich, UK), the vials were degassed with nitrogen and stored at −80°C prior to derivatization and GC-MS analysis.

Dried extracts were re-suspended in 500 µL of methanol. Aliquot of 200 µL was transferred to a vial and dried in a vacuum. The samples were derivatized with 40 µL of methoxyamine hydrochloride (MAHC, Sigma-Aldrich) (20 mg mL$^{-1}$) in pyridine (Sigma-Aldrich) for 90 min at 30°C at 150 rpm, and with 80 µL N-methyl-N-(trimethylsilyl) trifluoroacetamide with 1% trimethylchlorosilane (MSTFA with 1% TMCS, Thermo Scientific) for 120 min at 37°C at 150 rpm. Alkane series (10 µL, C10–C40, Supelco) in hexane (Sigma-Aldrich) and 100 µL of hexane was added to each sample before GC-MS analysis.

## Metabolite analysis by gas chromatography-mass spectrometry

The GC-MS system consisted of Agilent 7890A gas chromatograph with 7000 Triple quadruple mass spectrometer and GC PAL autosampler and injector (CTC Analytics). Splitless injection (1 µL) was employed using a deactivated single tapered splitless liner with glass wool (Topaz, 4 mm ID, Restek). Helium flow in the column (Agilent HP-5MS Ultra Inert, length 30 m, 0.25 mm ID, 0.25 µm film thickness combined with Agilent Ultimate Plus deactivated fused silica, length 5 m, 0.25 mm ID) was 1.2 mL min$^{-1}$ and purge flow at 0.60 min was 50 mL min$^{-1}$. The injection temperature was set to 270°C, MS interface 180°C, source 230°C and quadrupole 150°C. The oven temperature program was as follows: 2 min at 50°C, followed by a 7 °C min$^{-1}$ ramp to 260°C, 15 °C min$^{-1}$ ramp to 325°C, 4 min at 325°C and post-run at 50°C for 4.5 min. Mass spectra were collected with a scan range of 55–550 *m/z*.

Metabolite Detector (versions 2.06 beta and 2.2N) (*Hiller et al., 2009*) and AMDIS (version 2.68, NIST) were used for deconvolution, component detection and quantification. Malate levels were calculated as the peak area of the metabolite normalized with the peak area of the internal standard, glycerol-d$_8$, and the fresh weight of the sample.

## Measurements of NADPH-MDH activity

From light-adapted plants grown for 5 weeks (100–120 µmol m$^{-2}$ s$^{-1}$ at an 8 hour day photoperiod), total extracts were prepared as for non-aqueous fractionation in the extraction buffer supplemented with 250 µM DTT. In microplates, 5 µL of the extract (diluted x 500) were mixed with 20 µL of activation buffer, 0.1 M Tricine-KOH (pH 8.0), 180 mM KCl, 0.5% Triton X-100. Initial activity was measured immediately after, while total activity was measured after incubation for 2 hr at room temperature in presence of additional 150 mM DTT. Then assay mix was added consisting of 20 µL of assay buffer [0.5 M Tricine-KOH (pH 8.0), 0.25% Triton X-100, 0.5 mM EDTA], 9 µL of water, and 1 µL of 50 mM NADPH (prepared in 50 mM NaOH), after which 45 µL of 2.5 mM oxaloacetate or water control was added. The reaction was mixed, and light absorbance at 340 nm wavelength was measured at 25°C.

## Analysis of *rcd1* misregulated genes in microarray experiments related to chloroplast or mitochondrial dysfunction

Genes with misregulated expression in *rcd1* were selected from our previous microarray datasets (*Brosché et al., 2014*) with the cutoff, absolute value of logFC <0.5. These genes were subsequently clustered with the *rcd1* gene expression dataset together with various Affymetrix datasets related to chloroplast or mitochondrial dysfunction from the public domain using bootstrapped Bayesian hierarchical clustering as described in *Wrzaczek et al. (2010)*. Affymetrix raw data (.cel files) were normalized with Robust Multi-array Average normalization, and manually annotated to control and treatment conditions, or mutant *versus* wild type.

Affymetrix ATH1-121501 data were from the following sources: Gene Expression Omnibus https://www.ncbi.nlm.nih.gov/geo/, AA 3 hr (in figures labelled as experiment 1), GSE57140 (*Ivanova et al., 2014*); AA and H$_2$O$_2$, 3 hr treatments (in figures labelled as experiment 2), GSE41136 (*Ng et al., 2013b*); MV 3 hr, GSE41963 (*Sharma et al., 2013*); *mterf6-1*, GSE75824 (*Leister and Kleine, 2016*); *prors1-2*, GSE54573 (*Leister et al., 2014*); H$_2$O$_2$ 30 min, GSE43551 (*Gutiérrez et al., 2014*); high light 1 hr (in figures labelled as experiment 1), GSE46107 (*Van Aken et al., 2013*); high light 30 min in cell culture, GSE22671 (*González-Pérez et al., 2011*); high light 3

hr (in figures labelled as experiment 2), GSE7743 (*Kleine et al., 2007*); oligomycin 1 and 4 hr, GSE38965 (*Geisler et al., 2012*); norflurazon – 5 day-old seedlings grown on plates with norflurazon, GSE12887 (*Koussevitzky et al., 2007*); *msh1 recA3* double mutant, GSE19603 (*Shedge et al., 2010*). AtGenExpress oxidative time series, MV 12 and 24 hr, http://www.arabidopsis.org/servlets/TairObject?type=expression_set&id=1007966941. ArrayExpress, https://www.ebi.ac.uk/arrayexpress/: rotenone, 3 and 12 hr, E-MEXP-1797 (*Garmier et al., 2008*); *alx8* and *fry1*, E-MEXP-1495 (*Wilson et al., 2009*); *ndufs4*, E-MEXP-1967 (*Meyer et al., 2009*).

## Quantitative PCR

Quantitative PCR was performed essentially as described in *Brosché et al. (2014)*. The data were normalized with three reference genes, *PP2AA3*, *TIP41* and *YLS8*. Relative expression of the genes *RCD1*, *AOX1a*, *UPOX*, *ANAC013*, *At5G24640* and *ZAT12* was calculated in qBase +3.2 (Biogazelle, https://www.qbaseplus.com/). The primer sequences and primer efficiencies are presented in *Supplementary file 1*.

## Identification of interacting proteins using IP/MS-MS

Immunoprecipitation experiments were performed in three biological replicates as described previously (*De Rybel et al., 2013*), using 3 g of rosette leaves from p35S: ANAC013-GFP and 2.5 g of rosette leaves from pUBI10: RCD1-3xVenus transgenic lines. Interacting proteins were isolated by applying total protein extracts to αGFP-coupled magnetic beads (Milteny Biotech). Three replicates of p35S: ANAC013-GFP or pUBI10: RCD1-3xVenus were compared to three replicates of Col-0 controls. Tandem mass spectrometry (MS) and statistical analysis using MaxQuant and Perseus software was performed as described previously (*Wendrich et al., 2017*).

## HEK293T human embryonic kidney cell culture and transfection

HEK293T cells were maintained at 37°C and 5% $CO_2$ in Dulbecco's Modified Eagle's Medium F12-HAM, supplemented with 10% fetal bovine serum, 15 mM HEPES, and 1% penicillin/streptomycin. Cells were transiently transfected using GeneJuice (Novagen) according to the manufacturer's instructions.

For co-immunoprecipitation experiments, HEK293T cells were co-transfected with plasmids encoding HA-RCD1 and ANAC013-myc or ANAC017-myc. Forty hours after transfection, cells were lysed in TNE buffer [50 mM Tris-HCl (pH 7.4), 150 mM NaCl, 5 mM EDTA, 1% Triton X-100, 1 x protease inhibitor cocktail, 50 µM proteasome inhibitor MG132 (Sigma-Aldrich)]. After incubation for 2 hr at 4°C, lysates were cleared by centrifugation at 18 000 x *g* for 10 min at 4°C. For co-immunoprecipitation, cleared cell lysates were incubated with either αHA or αmyc antibody immobilized on agarose beads overnight at 4°C. Beads were washed six times with the lysis buffer. The bound proteins were dissolved in SDS sample buffer, resolved by SDS-PAGE, and immunoblotted with the specified antibodies.

## Protein expression and purification

The C-terminal domain of RCD1 for NMR study was expressed as GST-fusion protein in *E.coli* BL21 (DE3) Codon Plus strain and purified using GSH-Sepharose beads (GE Healthcare) according to the manufacturer's instruction. Cleavage of GST tag was performed with thrombin (GE Healthcare, 80 units per mL of beads) for 4 hr at room temperature and the C-terminal domain of RCD1 was eluted from the beads with PBS buffer (137 mM NaCl, 2.7 mM KCl, 10 mM $Na_2HPO_4$, 1.8 mM $KH_2PO_4$, pH 7.4). The protein was further purified by gel filtration with HiLoad 16/600 Superdex 75 column (GE Healthcare) equilibrated with 20 mM sodium phosphate buffer (pH 6.4), 50 mM NaCl at 4°C.

## Peptide synthesis

ANAC013 peptides of >98% purity for surface plasmon resonance and NMR analysis were purchased from Genecust, dissolved in water to 5 mM final concentration and stored at −80°C before analyses.

## Surface Plasmon resonance

The C-terminal domain of RCD1 was covalently coupled to a Biacore CM5 sensor chip *via* amino-groups. 500 nM of ANAC013 peptides were then profiled at a flow rate of 30 µL min$^{-1}$ for 300 s, followed by 600 s flow of running buffer. Analysis was performed at 25°C in the running buffer containing 10 mM HEPES (pH 7.4), 150 mM NaCl, 3 mM EDTA, 0.05% surfactant P20 (Tween-20). After analysis in BIAevaluation (Biacore) software, the normalized resonance units were plotted over time with the assumption of one-to-one binding.

## NMR spectroscopy

NMR sample production and chemical shift assignment have been described in *Tossavainen et al. (2017)*. A Bruker Avance III HD 800 MHz spectrometer equipped with a TCI $^1$H/$^{13}$C/ $^{15}$N cryoprobe was used to acquire spectra for structure determination of RCD1$^{468-589}$. Peaks were manually picked from three NOE spectra, a $^1$H, $^{15}$N NOESY-HSQC and $^1$H, $^{13}$C NOESY-HSQC spectra for the aliphatic and aromatic $^{13}$C regions. CYANA 2.1 (*López-Méndez and Güntert, 2006*) automatic NOE peak assignment – structure calculation routine was used to generate 300 structures from which 30 were further refined in explicit water with AMBER 16 (*Case et al., 2005*). Assignments of three NOE peaks were kept fixed using the KEEP subroutine in CYANA. These NOE peaks restrained distances between the side chains of W507 and M508 and adjacent helices 1 and 4, respectively. Fifteen lowest AMBER energy structures were chosen to represent of RCD1$^{468-589}$ structure in solution.

Peptide binding experiment was carried out by preparing a sample containing RCD1$^{468-589}$ and ANAC013$^{235-284}$ peptides in an approximately 1:2 concentration ratio, and recording a $^1$H, $^{15}$N HSQC spectrum. Amide peak positions were compared with those of the free RCD1$^{468-589}$.

## Acknowledgements

We thank Dr. Olga Blokhina, Dr. Bernadette Gehl and Manuel Saornil for the help in studies of mitochondria, Katariina Vuorinen for genotyping *rcd1-1 anac017*, Richard Gossens for critical comments on the manuscript, Prof. Francisco Javier Cejudo (Institute of Plant Biochemistry, University of Sevilla) for the α2-CP antibody, and Dr. Saijaliisa Kangasjärvi for the αSOT12 antibody. We acknowledge CSC – IT Center for Science, Finland, for computational resources. This work was supported by the University of Helsinki (JK); the Academy of Finland Centre of Excellence programs (2006–11; JK and 2014–19; JK, EMA) and Research Grant (Decision 250336; JK); Academy of Finland fellowships (Decisions 275632, 283139 and 312498; MW); Academy of Finland Research Grant (Decision 288235; PP); by The Research Foundation – Flanders (FWO; Odysseus II G0D0515N and Post-doc grant 12D1815N; BW and BDR); PlantaSYST project by the European Union's Horizon 2020 research and innovation programme (SGA-CSA No 664621 and No. 739582 under FPA No. 664620; SA and ARF); Deutsche Forschungsgemeinschaft (DFG TRR 175 The Green Hub – Central Coordinator of Acclimation in Plants; FA and ARF); the Research Foundation – Flanders (Excellence of Science project no 30829584; FVB).

## Additional information

### Funding

| Funder | Grant reference number | Author |
| --- | --- | --- |
| Deutsche Forschungsgemeinschaft | DFG TRR 175 | Fayezeh Aarabi<br>Alisdair R Fernie |
| Horizon 2020 Framework Programme | FPA No. 664620 | Saleh Alseekh<br>Alisdair R Fernie |
| Fonds Wetenschappelijk Onderzoek | Odysseus II G0D0515N | Brecht Wybouw<br>Bert De Rybel |
| Fonds Wetenschappelijk Onderzoek | 12D1815N | Brecht Wybouw<br>Bert De Rybel |

| Fonds Wetenschappelijk On-derzoek | 30829584 | Frank Van Breusegem |
| --- | --- | --- |
| Suomen Akatemia | 288235 | Perttu Permi |
| Suomen Akatemia | Centre of Excellence Program 2014–19 | Eva-Mari Aro<br>Jaakko Kangasjärvi |
| Suomen Akatemia | 275632 | Michael Wrzaczek |
| Suomen Akatemia | 283139 | Michael Wrzaczek |
| Suomen Akatemia | 312498 | Michael Wrzaczek |
| Suomen Akatemia | Centre of Excellence Program 2006–11 | Jaakko Kangasjärvi |
| Helsingin Yliopisto | | Jaakko Kangasjärvi |
| Suomen Akatemia | 250336 | Jaakko Kangasjärvi |

The funders had no role in study design, data collection and interpretation, or the decision to submit the work for publication.

### Author contributions

Alexey Shapiguzov, Conceptualization, Resources, Formal analysis, Validation, Investigation, Visualization, Methodology, Writing—original draft, Project administration, Writing—review and editing; Julia P Vainonen, Conceptualization, Validation, Investigation, Methodology, Writing—original draft, Writing—review and editing; Kerri Hunter, Resources, Investigation, Methodology; Helena Tossavainen, Data curation, Formal analysis, Validation, Investigation, Methodology; Arjun Tiwari, Maarit Hellman, Brecht Wybouw, Validation, Investigation, Methodology; Sari Järvi, Conceptualization, Resources, Validation, Investigation, Methodology, Project administration; Fayezeh Aarabi, Saleh Alseekh, Katrien Van Der Kelen, Lauri Nikkanen, Nina Sipari, Investigation, Methodology; Julia Krasensky-Wrzaczek, Conceptualization, Writing—original draft, Writing—review and editing; Markku Keinänen, Conceptualization, Supervision, Writing—review and editing; Esa Tyystjärvi, Conceptualization, Writing—review and editing; Eevi Rintamäki, Conceptualization, Validation; Bert De Rybel, Funding acquisition, Validation, Investigation, Methodology; Jarkko Salojärvi, Data curation, Software, Formal analysis, Validation, Investigation, Methodology; Frank Van Breusegem, Perttu Permi, Conceptualization, Supervision, Funding acquisition, Validation, Methodology, Project administration; Alisdair R Fernie, Conceptualization, Supervision, Funding acquisition, Validation, Methodology, Project administration, Writing—review and editing; Mikael Brosché, Conceptualization, Formal analysis, Investigation, Methodology, Writing—review and editing; Eva-Mari Aro, Conceptualization, Resources, Funding acquisition, Project administration; Michael Wrzaczek, Conceptualization, Resources, Supervision, Funding acquisition, Validation, Methodology, Project administration, Writing—review and editing; Jaakko Kangasjärvi, Conceptualization, Resources, Supervision, Funding acquisition, Validation, Visualization, Methodology, Writing—original draft, Project administration, Writing—review and editing

### Author ORCIDs

Alexey Shapiguzov (ID) https://orcid.org/0000-0001-7199-1882
Kerri Hunter (ID) https://orcid.org/0000-0002-2285-6999
Saleh Alseekh (ID) http://orcid.org/0000-0003-2067-5235
Brecht Wybouw (ID) http://orcid.org/0000-0001-8783-4646
Julia Krasensky-Wrzaczek (ID) https://orcid.org/0000-0002-5989-9984
Bert De Rybel (ID) http://orcid.org/0000-0002-9551-042X
Jarkko Salojärvi (ID) https://orcid.org/0000-0002-4096-6278
Michael Wrzaczek (ID) https://orcid.org/0000-0002-5946-9060
Jaakko Kangasjärvi (ID) http://orcid.org/0000-0002-8959-1809

### Decision letter and Author response

Decision letter https://doi.org/10.7554/eLife.43284.079
Author response https://doi.org/10.7554/eLife.43284.080

# Additional files

## Supplementary files

• Supplementary file 1. Primers used in the study.

DOI: https://doi.org/10.7554/eLife.43284.044

• Transparent reporting form

DOI: https://doi.org/10.7554/eLife.43284.045

## Data availability

The atomic coordinates and structural restraints for the C-terminal domain of RCD1 have been deposited in the Protein Data Bank with the accession code 5N9Q.

The following dataset was generated:

| Author(s) | Year | Dataset title | Dataset URL | Database and Identifier |
|---|---|---|---|---|
| Tossavainen H, Hellman M, Vaino-nen JP, Kangasjärvi J, Permi P | 2017 | 1H, 13C and 15N NMR chemical shift assignments of A. thaliana RCD1 RST | https://pdbj.org/mine/summary/5n9q | Protein Data Bank Japan, 5N9Q |

The following previously published datasets were used:

| Author(s) | Year | Dataset title | Dataset URL | Database and Identifier |
|---|---|---|---|---|
| Ivanova A, Law SR, Narsai R, Duncan O, Hoon J, Zhang B | 2014 | Expression data of Col:LUC Arabidopsis treated with antimycin A (AA) in the presence or absence of a synthetic auxin analogue | https://https://www.ncbi.nlm.nih.gov/geo/query/acc.cgi?acc=GSE57140 | NCBI Gene Expression Omnibus, GSE57140 |
| Ng S, Ivanova A, Duncan O, Law SR, Van Aken O, De Clercq I, Wang Y, Carrie C, Xu L, Kmiec B, Walker H, Van Breusegem F, Whelan J, Giraud E | 2013 | A membrane-bound NAC transcription factor, ANAC017, mediates mitochondrial retrograde signaling in Arabidopsis | https://www.ncbi.nlm.nih.gov/geo/query/acc.cgi?acc=GSE41136 | NCBI Gene Expression Omnibus, GSE41136 |
| Raghvendra S | 2013 | Gene expression analysis in wild-type and OsGRX8 overexpression line in response to various treatments | https://www.ncbi.nlm.nih.gov/geo/query/acc.cgi?acc=GSE41963 | NCBI Gene Expression Omnibus, GSE41963 |
| Kleine T, Leister D | 2015 | Expression data from pam48 (mterf6-1) mutants | https://www.ncbi.nlm.nih.gov/geo/query/acc.cgi?acc=GSE75824 | NCBI Gene Expression Omnibus, GSE75824 |
| Kleine T, Leister D | 2014 | Identification of target genes of translation-dependent signalling in Arabidopsis | https://www.ncbi.nlm.nih.gov/geo/query/acc.cgi?acc=GSE54573 | NCBI Gene Expression Omnibus, GSE54573 |
| Gutiérrez J, Gonzá-lez-Pérez S, García-García F, Lorenzo O, Revuelta JL, McCabe PF, Arella-no JB | 2014 | Chloroplast-dependent programmed cell death is activated in Arabidopsis cell cultures after singlet oxygen production by Rose Bengal | https://www.ncbi.nlm.nih.gov/geo/query/acc.cgi?acc=GSE43551 | NCBI Gene Expression Omnibus, GSE43551 |
| Narsai R | 2013 | Expression data in response to WRKY40 and WRKY63 knock-out/overexpression (and in response to high light stress) | https://www.ncbi.nlm.nih.gov/geo/query/acc.cgi?acc=GSE46107 | NCBI Gene Expression Omnibus, GSE46107 |
| Arellano JB, Dopa-zo J, García-García F, González-Pérez S, Lorenzo O, Osuna D, Revuelta JL | 2010 | Gene expression from Arabidopsis under high light conditions | https://www.ncbi.nlm.nih.gov/geo/query/acc.cgi?acc=GSE22671 | NCBI Gene Expression Omnibus, GSE22671 |
| Strand A, Kleine T, | 2007 | Genome-wide gene expression | https://www.ncbi.nlm. | NCBI Gene |

| | | | | |
|---|---|---|---|---|
| Kindgren P, Benedict C, Hendrickson L | | analysis reveals a critical role for CRY1 in the Response of Arabidopsis to High Irradiance | nih.gov/geo/query/acc.cgi?acc=GSE7743 | Expression Omnibus, GSE7743 |
| Geisler DA, Päpke C, Persson S | 2012 | Effect of oligomycin on transcript levels in Arabidopsis seedling cultures | https://www.ncbi.nlm.nih.gov/geo/query/acc.cgi?acc=GSE38965 | NCBI Gene Expression Omnibus, GSE38965 |
| Nott A, Koussevitzky S, Mockler T, Mockler T, Hong F, Chory J | 2008 | Differential response of gun mutants to norflurazon | https://www.ncbi.nlm.nih.gov/geo/query/acc.cgi?acc=GSE12887 | NCBI Gene Expression Omnibus, GSE12887 |
| Shedge V | 2009 | Expression data from Arabipdosis msh1 recA3 double mutant under heat stress | https://www.ncbi.nlm.nih.gov/geo/query/acc.cgi?acc=GSE19603 | NCBI Gene Expression Omnibus, GSE19603 |
| Delannoy E | 2008 | Transcription profiling by array of Arabidopsis cell cultures treated with rotenone | https://www.ebi.ac.uk/arrayexpress/experiments/E-MEXP-1797/ | ArrayExpress Archive of Functional Genomics Data, E-MEXP-1797 |
| Wilson PB | 2009 | Transcription profiling of Arabidopsis wild type and SAL1 mutant plants grown under normal | https://www.ebi.ac.uk/arrayexpress/experiments/E-MEXP-1495/ | ArrayExpress, E-MEXP-1495 |
| Meyer EH | 2008 | Transcription profiling of Arabidopsis wild type and complex I mutant plants | https://www.ebi.ac.uk/arrayexpress/experiments/E-MEXP-1967/ | ArrayExpress, E-MEXP-1967 |

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
