## [Decision Letter]

[Editors’ note: a previous version of this study was rejected after peer review, but the authors submitted for reconsideration. The first decision letter after peer review is shown below.]

Thank you for submitting your work entitled "RCD1 Coordinates Chloroplastic and Mitochondrial Electron Transfer through Interaction with ANAC Transcription Factors" for consideration by *eLife*. Your article has been reviewed by two peer reviewers, and the evaluation has been overseen by a Reviewing Editor and a Senior Editor.

Our decision has been reached after consultation between the reviewers. Based on these discussions and the individual reviews below, we regret to inform you that your work will not be considered for publication in *eLife* in its present form.

However, we would be ready to have a second look at a resubmission of a substantially revised manuscript that comprehensively addresses the reviewer comments, including the following key experiments:

1) One important experiment would be to look for suppression of NAC17-dependent retrograde signalling in an RCD overexpression and/or RCD 7cys complemented line. Notably, these lines should be readily available, and as such these are not very challenging experiments.

2) It must be clarified whether the initial MDH activity was indeed measured after 2 h of incubation in the absence of DTT, as it presently reads in subsection “Measurements of NADPH-MDH activity”. This would have caused inactivation. If the non-aqueous fractions still exist, chloroplast malate levels would be most interesting rather than whole leaf amounts.

3) Finally, the manuscript should be substantially rewritten and rearranged. In particular, you would have to streamline your paper and focus on the novel aspects. Confirmatory experiments with respect to the literature should receive less attention and be displayed in the extended data. Moreover, often the results can only be understood when investigating the figure, reading the legend, the corresponding section in the Materials and methods and the text passage. You should aim to ease this process. For example you could provide the most important information in the legend, e.g. the light intensity used for NADPH-MDH-activity measurement (Figure 3), the incubation time with MV in Figure 3B or the positions of standard polypeptides in the Western blots.

Reviewer #1:

This paper reports on a complex but highly interesting story, based on the compilation of diverse data linking RADICAL CELL DEATH 1(RCD1) function with regulation of redox and reactive oxygen species signaling and brings this in the wider context of retrograde information exchange in photosynthetic cells. The effect of lacking RCD1 on AOX transcript accumulation is highly interesting and correlated with high accumulation of AOX1A. Inefficient ATP production by the respiratory chain in the mutants was compensated by increased activity of glycolysis and linked pathways such as oxidative pentose pathway.

The strength of the study is that the authors attempt to test their hypotheses by using specific mutants. Another important point is the establishment of the link between the RCD1-dependent signaling and ANAC13/ 17, physically (shown before) and now deepened for the proteins ANAC13/ ANAC17/ SRO1/ RCD1 (also in human embryonic kidney cell; these results could move to the supplementary materials), indirectly functionally by corresponding changes in the transcriptomes and by increased ROS sensitivity of rd1/anac17 double mutants. The structural part on the interaction between RCD1 and ANAC13/ 17 is also significant.

It has to be taken with caution that H_2_O_2_ and methylviologen treatments specifically mimic chloroplast ROS production. This must be discussed with more care, also in the Discussion final paragraph. Exogenous application of H_2_O_2_ acts in different cells and cell compartments, likewise MV supplementation is not a specific inducer of chloroplast ROS. Thus high light treatment affects analyzed parameters different to H_2_O_2_ and methylviologen.

Figure 3: Subsection “Evidence for increased electron transfer between chloroplasts and mitochondria in rcd1”: The activation state of NADPH malate dehydrogenase (MDH) is very low, almost unbelievably low (4-8%). Doubling the activity from this very low activation state unlikely is sufficient to explain any changes in rcd1 mutants. Kinetic modelling could help to define whether the observation is significant or not. The legend to the figure is insufficient to understand what is measured. Describe the photosynthetic condition the leaves were exposed to here. The concern with this interpretation is strengthened by the unchanged NADPH/NADP-ratio in non-aqueously isolated chloroplasts. The incubation of the test with and without DTT for 2 h at room temperature is not clear: Was the minus DTT-sample supposed to be the initial activity? During a 2 h incubation with 250 μM DTT, the initially active MDH may inactivate and this may explain the extremely low activity? This ambiguity must be clarified. Possibly the measurements would have to be repeated if they are considered to be central to the paper.

The comfort of reading is not always optimal at present since the reader must simultaneously consult the text, the legend and the method.

The high concentrations of NADH and NADPH in the vacuole appears surprising and an artefact. The description of the method and results may not be complete here. Which other compartment markers were measured and included in the best fit? May be it would be more reliable and convincing to present total leaf nucleotide levels rather than somewhat questionable date on subcellular compartments. The statistically significant difference in vacuolar NADH points to severe problems with this method.

The discussion is mostly fine, but sometimes quite speculative. Thus the scenario described for ROS-mediated aggregation of RCD1 and subsequent release of RCD1-dependent repression may still be considered as hypothesis. Were the ANAC13/ 17 targets expressed in the lines complemented with the heptuple Cys-to-Ser variant upon oxidative treatment?

Despite these points, I consider this submission as an important contribution to the field, which should attract attention of the wider readership.

Reviewer #2:

In this study, Shapigzov, Vainonen et al. further explore the role of RCD1 (previously picked up in an ozone sensitivity screen) in regulation of electron transfer and retrograde signalling. The manuscript represents a vast amount of work and provides interesting findings in many aspects of stress response and signalling. The main novel point the study puts forward is that RCD1 may repress (mitochondrial) retrograde signalling via direct binding to ANAC013/17 transcription factors, and repressing their function. This system may be of importance during resistance to the ROS-generating herbicide methyl viologen (paraquat, leading to PSI superoxide and H_2_O_2_ production).

The study first confirms MV resistance of the *rcd1* mutant reported previously (e.g. Fujibe et al., 2004, Hiltscher et al., 2014). The authors show that *rcd1* mutants maintain Fv/Fm much better after light exposure in the presence of MV, compared to WT, but that this is not just because MV cannot reach PSI. This is accompanied by lower H_2_O_2_ production in *rcd1*, indicating that somehow loss of RCD1 (a nuclear protein) suppresses ROS formation by MV. The observation that 2-cys peroxiredoxin is in a more reduced state therefore makes sense as less H_2_O_2_ seems to be present in the *rcd1* cells.

The study then shifts focus to mitochondrial alternative respiration (AOX-dependent). AOX1a gene expression was previously found by this group to be induced in *rcd1* mutants, and this is further confirmed at the protein level and by increased KCN-resistant respiration. This indicates that the increased alternative respiration in *rcd1* is largely due to AOX1a activity. The authors further indicate an increased flux through e.g. glycolysis, indicating a different metabolic state, likely independent of AOX1a.

Pre-treatment with mitochondrial complex III inhibitor AA could partially mimic the *rcd1* resistance to MV in WT plants, indicating that mitochondrial retrograde signalling can be of importance in MV tolerance, which is very interesting. Inhibition of AOX by SHAM made the *rcd1* less MV tolerant, indicating the AOX is needed for full MV tolerance. This is one of the most interesting findings in the study, that AOX is part of the *rcd1* mechanism for MV resistance. Follow-up experiments however showed that overexpression of AOX1a itself was not sufficient to induce MV tolerance, and *rcd1* aox1a plants were as MV resistant as *rcd1*. This suggests that other isoforms than AOX1a may be important here, and that higher AOX capacity is not sufficient in itself, and there might be additional post-translational regulation. The point of the experiment shown in Figure 3A may not be very clear to the general reader, and needs better explanation.

The authors then check NADPH-MDH activity and malate content. Both are higher in *rcd1* than in Col-0 after MV treatment. Overall NADPH-MDH capacity was higher in *rcd1*, suggesting some flux of reductive equivalent to the mitochondria via chloroplastic malate formation. The increased activation state may again have to do with the lower ROS oxidation state of the *rcd1* cells.

A transcriptome meta-analysis further confirms the previous findings that many mitochondrial retrograde regulation target genes are constitutively induced in *rcd1*, as well as by MV.

The authors then make a link to another retrograde pathway mediated by PAP, which also co-regulates MDS gene expression (Van Aken and Pogson, 2017). The authors cross *rcd1* with a PAP accumulating mutant *sal1* and find a much stronger growth inhibition phenotype than either of the single mutants. Although this quite interesting in itself, the authors don't really provide a mechanistic explanation or suggestion for this additive phenotype. The Sal1 and *rcd1 sal1* mutants would need to be analysed in more detail (e.g. Fv/Fm MV analysis) to show PAP is really involved here.

Constitutive induction of MDS transcripts (including ANAC013) genes suggests that their main positive transcription factors are activated (ANAC017/013). To address this, the authors made the interesting cross of *rcd1* anac017. Indeed, AOX levels and capacity are lower in this double mutant, supporting the role of ANAC017 in MDS upregulation in *rcd1*. The anac017 mutant was previously shown to be highly MV-sensitive, and ANAC013/ANAC017 overexpression lines are more tolerant (e.g. De Clercq et al., 2013). The observed hypersensitivity of the anac017 mutant in the Fv/Fm MV assay is in line with this. The *rcd1* anac017 is more sensitive than *rcd1* indicating the NAC MRR pathway plays a partial role in the *rcd1* MV phenotype. No descriptions of the visual, developmental and stress (e.g. ozone) phenotype of the *rcd1 anac017* mutant are provided, which would be very interesting.

The authors then support the previously reported interaction of ANAC013 with RCD1 (Jaspers et al., 2009, O'Shea et al., 2017) using Co-IP strategies, and provide further insight into the interaction with surface plasmon resonance and NMR.

Finally, the authors show that RCD1 protein level (or at least HA-RCD1) is degraded over time by MV and H_2_O_2_ treatment. Using non-reducing gels, they show that quickly after these treatments, a more oxidised multimeric form of RCD1 is found, indicating protein aggregation. From my interpretation of Figure 6B and Supp S5B, H_2_O_2_ makes a predominantly much higher MW aggregate than MV, so this should be discussed. By mutating 7 Cys residues in the RCD1 protein, the authors showed that this aggregation can be largely blocked. I am somewhat confused by the statement on mitochondrial inhibitor AA in the second paragraph of subsection “RCD1 protein is sensitive to organellar ROS” and Figure S5B. To me the total (reducing gel) abundance of RCD1 after AA is much lower than after MV. This seems in contrast to the statement in the text that RCD1 is not a direct target of a mitochondrial signal from AA. This should be clarified. Also H_2_O_2_ seems much stronger in affecting RCD1 abundance than MV. In line with this, MV is a much weaker inducer of MDS genes than AA or H_2_O_2_ (Figure 4A), so perhaps the role of RCD1 is of more importance in AA signalling than in MV, but this is under explored. In S5C, I assume the legend is mislabelled, where Col-0 should be the grey line? The identical phenotype of the 7cys RCD1 complementation compared to the normal RCD1 complementation, however questions the physiological importance of the RCD1 redox-induced aggregation, as one might expect then less efficient complementation.

Overall, the manuscript contains lots of data, and the experiments seem well performed so I trust their validity. To a large extent, many of the findings are confirming previous findings from the large body of literature surround RCD1 and MRR (binding with ANACs, upregulation of multiple MDS genes also in PAP-related mutants, MV tolerance phenotypes, potential involvement of AOX1a, involvement of ANAC013/17 in MV tolerance), and expand on them with additional techniques and measurements. The main model proposed here is that RCD1 binds and inhibits ANAC017/13, so mitochondrial retrograde signalling is blocked when no stress is present. Upon stress, the ROS-induced redox/changes alter RCD1 configuration/abundance and cause aggregation, which may reduce the inhibitory capacity of RCD1 on the NACs, thereby inducing MDS gene expression. This system is of importance also for chloroplast stress resistance such as MV, involving AOX activity.

If this is indeed true, that would be a major finding in the field, providing mechanistic insight into the relatively poorly understood ANAC MRR pathway. However, at this stage the evidence brought here is fairly circumstantial, with many nice experiments that however do not provide conclusive evidence. Many of the key findings that lead to this conclusion are already published (RCD1 ANAC interaction, induction of ANAC target genes in *rcd1* mutants, role of ANACs in MV resistance). Indeed, the interactions have now been confirmed by several labs independently and with great detail, so it seems very likely to occur. However, many mutants/treatments have activation of MDS genes as reported in many studies, but this may just indicate a basally stressed state of the plants (not-necessarily just ROS levels) leading to ANAC signalling, and not necessarily direct regulation on the transcription factor.

To make this more convincing, more direct experiments should be performed. One would for instance expect that an RCD1 Overexpression line (which has been made in the past, Fujibe 2006, Biosci. Biotechnol. Biochem., 70 (8), 1827-1831, and complements the *rcd1* MV resistance phenotype) would suppress (AA-induced) MRR to a significant extent. The 7cys RCD1 line would also be of interest in this context, to show that redox state can affect retrograde signalling. A complementation line of mutated RCD1 lacking the clearly defined ANAC interaction domain should not be able to restore wild-type levels of MV susceptibility and MDS upregulation, but I am not aware if one is available. Experiments showing a dynamic release of sequestration/binding of ANACs from RCD1 during stress would be even more convincing, but I agree these are technically very challenging.

Overall, the study provides many new insights and is well written, and soundly presented. To really make a convincing case for their exciting model, some more specific experiments targeting the ANAC/RCD1 regulation would be beneficial.

[Editors’ note: what now follows is the decision letter after the authors submitted for further consideration.]

Thank you for resubmitting your work entitled "Arabidopsis RCD1 Coordinates Chloroplast and Mitochondrial Functions through Interaction with ANAC Transcription Factors" for further consideration at *eLife*. Your revised article has been favorably evaluated by Christian Hardtke (Senior Editor), a Reviewing Editor, and two reviewers.

The manuscript has been improved, but the reviewers are of the opinion that you did not adequately respond to key requests from the first round of reviews. We would thus like to ask you to carefully consider the reviews pasted below, and revise your manuscript accordingly.

*Reviewer #1:*

In this revised version the authors addressed many small and large issues raised by the reviewers and editor. However, I feel some of the main concerns have not been addressed appropriately yet.

The authors try to introduce a new mechanism of how RCD1 directly binds and inhibits ANAC017/13. This possibly interaction represses the transcription factors and thus would keep MRR target gene expression under control, such as AOX1a. Upon oxidative stress, RCD1 aggregates and would then release some of the ANAC017/13 to induce MRR gene expression. This interesting model has a number of testable implications, which were suggested after initial review, and were also underlined to be important by the Editor. However some of these have not been addressed directly.

1) In comment by *eLife* Senior Editor: "One important experiment would be to look for suppression of NAC17-dependent retrograde signalling in an RCD overexpression and/or RCD 7cys complemented line. Notably, these lines should be readily available, and as such these are not very challenging experiments."

And reviewer 2 "To make this more convincing, more direct experiments should be performed. One would for instance expect that an RCD1 Overexpression line (which has been made in the past, Fujibe 2006, Biosci. Biotechnol. Biochem., 70 (8), 1827-1831, and complements the *rcd1* MV resistance phenotype) would suppress (AA-induced) MRR to a significant extent. The 7cys RCD1 line would also be of interest in this context, to show that redox state can affect retrograde signalling."

The authors respond in point 1 that a number of *rcd1* RCD1-HA with higher and lower levels of RCD1 (thus probably equivalent to an RCD1 overexpression line, but hard to say without RCD1 specific antibody, and only an anti-HA antibody) have been generated. The authors then show the MV-dependent Fv/Fm phenotype is reverting to different extents to Col-0. This is nice, but the key experiment to address the raised question is then missing. When treated with a Mitochondrial Retrograde Regulation inducing agent (classically antimycin A), do these “overexpression” lines show a reduction in ANAC017-dependent MRR gene expression (e.g. Aox1a, UPOX, UGT74E2)? The authors did experiments along these lines but using MV (Figure 2—figure supplement 1D). MV however does not directly induce mitochondrial signalling as it mainly affects chloroplastic ROS, which is supported by the fact that MRR marker genes AOX1a and UPOX are not induced by MV (Figure 2—figure supplement 1D). I don't see why the authors did not do the same experiment using AA, which would address the question, and is relatively easy to perform. In the RCD1d7cys complemented line, which fails to aggregate, such a repression of AA-induction of MRR genes may not occur, and would thus be expected to show more WT-like gene expression responses.

2) A second experiment that was suggested was to use *rcd1* mutants complemented with RCD1 lacking the ANAC017 interaction domain (RST domain). In point 3 the authors describe RCD1dRST-HA complementation lines were generated. I assume this is indeed transformation into the *rcd1* mutant background, so I suggest these line are called *rcd1* RC1dRST-HA in the text and figures (e.g. Figure 7B-C), same for *rcd1* RCD1-HA. This to avoid confusion that they are in the Col-0 background, which would lead to a different interpretation. The results in Figure 7B and C are indeed in line with the fact that the RST1 domain is important for RCD1 function in this context. As RCD1 is known to interact with a wide range of transcription factors, so it is not direct evidence for a role via interaction with ANAC017/13, but overall that is a nice result.

3) My comments regarding the size of the aggregates have been addressed.

4) Also the comments regarding Fv/Fm measurement on the *rcd1 sal1* have been addressed, it is interesting to see that it shows much higher tolerance to MV than *rcd1* itself.

5) I still think it is important to show the visual phenotype of the *rcd1 anac017* mutant plants, even if they don't seem to revert the *rcd1* ozone resistance phenotype, as is partially observed for the MV Fv/Fm phenotype. This makes some sense, as *anac017* is not known to be involved in ozone tolerance.

6) I don't really understand the author's reasoning in point 37 of their reply letter. The authors say there is no effect of AA on RCD1-HA redox state, but Figure 4—figure supplement 2B to me shows induction of higher order complexes by AA and myx of the same size as MV in light or H_2_O_2_, albeit weaker. 2.5 μm is a relatively low concentration, as 50 μm AA has been used most extensively throughout the literature. They also say that because MDS gene induction (by which they mean AOX protein level changes using Western blots) by AA is still occurring in *rcd1*, this leads them to conclude that RCD1 is unlikely part of AA signalling. As RCD1 is suggested to suppress ANAC017 signalling, an *rcd1* KO should still show AA gene induction, as the potential repressor is now inactive. An RCD1 OX line may display reduction in AA signalling, but this question has still not been addressed, see Point 1.

Reviewer #2:

The authors have thoroughly revised their previous manuscript version and provide additional evidence for their lines of thoughts and conclusions. The study contributes important and novel information to our understanding of retrograde signaling and the integration of chloroplast and mitochondrial retrograde signals. In the following you find the combined list of minor and very few more critical points.

Introduction section, first paragraph: “… transport of carbon rich metabolites…” It is not really clear what is meant? Carbon rich might be a fatty acid? energy-rich like dihydroxyacetone phosphate or malate?

Introduction section, second paragraph: “… leakage of electrons…”; is this still an adequate description of the process? Our concept of ROS signaling more and more assumes that ROS are generated to enable regulation.

Introduction section, fourth paragraph: “… provides positive feedback regulation”; please be more precise and give the direction:.… provides positive feedback and thus enhancement of the signal.

Subsection “Mitochondrial respiration is altered in *rcd1*”, opening sentence: I would omit the U here, since it is confusing for the non-expert, and in the legend and Materials and methods it will be mentioned. But in fact the uniformity of label is not explained, is it?

Subsection “Retrograde signaling from both chloroplasts and mitochondria is altered in *rcd1*”: msh1, recA3 (add comma)

Discussion third: In the same paragraph: “suggests that RCD1 is not a redox sensor of mitochondrial ROS/ redox signaling” still sounds like an overinterpretation of the data and should be toned down. “unlikely is.…” In fact in context with the subsequent discussion, it is not really clear what this statement means. If mitochondrial alterations in metabolism cause redox changes in the nucleus/cytosol similar to those of the chloroplast, then the conformational changes should be the same. The power of mitochondrial redox/ROS signals may just be less than chloroplast redox/ROS signals.

Subsection “RCD1 regulates stress responses and cell fate”, first paragraph: Rap2.4a

---

## [Author Response]

[Editors’ note: the author responses to the first round of peer review follow.]

1) One important experiment would be to look for suppression of NAC17-dependent retrograde signalling in an RCD overexpression and/or RCD 7cys complemented line. Notably, these lines should be readily available, and as such these are not very challenging experiments.

1) To show dependence on *rcd1* phenotypes on abundance of expressed RCD1, we included new results on MV tolerance in the *rcd1* complementation lines expressing different levels of RCD1-HA (Figure 1—figure supplement 1). These lines were used instead of the RCD1-overexpressor line reported in (Fujibe, T. et al., 2006), because the latter line has raised concerns about possible induced silencing of *RCD1* or other members of the *SRO* gene family.

2) The lines expressing RCD1Δ7Cys-HA have now been described in more detail. New data has been added on MV tolerance and expression of MDS genes in these lines (Figure 2—figure supplement 1C, D). in vivoabundance of RCD1Δ7Cys-HA in comparison to RCD1-HA has been addressed (Figure 2—figure supplement 1C).

3) New data on RCD1ΔRST-HA complementation lines was generated and added. These lines express an RCD1 variant lacking the NAC interaction domain. AOX abundance and MV tolerance have been assessed in these lines (Figure 7B, C).

2) It must be clarified whether the initial MDH activity was indeed measured after 2 h of incubation in the absence of DTT, as it presently reads in subsection “Measurements of NADPH-MDH activity”. This would have caused inactivation. If the non-aqueous fractions still exist, chloroplast malate levels would be most interesting rather than whole leaf amounts.

4) The required information has been added to the Figure 5C legend and to Materials and methods. The initial NADPH-MDH activity was measured directly following the extraction. The 2-hour incubation was only performed in presence of 150 mM DTT prior to the measurement of the total NADPH-MDH activity.

5) The distribution of malate was measured in non-aqueous fractions isolated from Col-0 and *rcd1*. The data was included as Figure 5—figure supplement 2.

3) Finally, the manuscript should be substantially rewritten and rearranged. In particular, you would have to streamline your paper and focus on the novel aspects. Confirmatory experiments with respect to the literature should receive less attention and be displayed in the extended data. Moreover, often the results can only be understood when investigating the figure, reading the legend, the corresponding section in the Materials and methods and the text passage. You should aim to ease this process. For example you could provide the most important information in the legend, e.g. the light intensity used for NADPH-MDH-activity measurement (Figure 3), the incubation time with MV in Figure 3B or the positions of standard polypeptides in the Western blots.

6) The manuscript has been profoundly revised. In particular, the Results, the figure legends and the supplements have been entirely rearranged and rewritten to facilitate the reading.

7) The description of the experiments that were not directly related to the central message of the manuscript, or the confirmatory experiments, was shifted to the supplemental figure legends.

8) Comprehensibility of the figures was increased by including necessary explanations and experimental details in the figures and figure legends. Positions of molecular weight markers were added to immunoblot images where necessary. All figures were graphically redesigned to facilitate visual display of the data.

Reviewer #1:The strength of the study is that the authors attempt to test their hypotheses by using specific mutants. Another important point is the establishment of the link between the RCD1-dependent signaling and ANAC13/ 17, physically (shown before) and now deepened for the proteins ANAC13/ ANAC17/ SRO1/ RCD1 (also in human embryonic kidney cell; these results could move to the supplementary materials), indirectly functionally by corresponding changes in the transcriptomes and by increased ROS sensitivity of rd1/anac17 double mutants. The structural part on the interaction between RCD1 and ANAC13/ 17 is also significant.

*It has to be taken with caution that H_2_O_2_and methylviologen treatments specifically mimic chloroplast ROS production. This must be discussed with more care, also in the Discussion final paragraph. Exogenous application of H_2_O_2_acts in different cells and cell compartments, likewise MV supplementation is not a specific inducer of chloroplast ROS. Thus high light treatment affects analyzed parameters different to H_2_O_2_ and methylviologen.*

This concern has been addressed by adding a data panel showing that MV only triggered RCD1 oligomerization under light, but not in the darkness (Figure 4—figure supplement 2B). This observation hints on chloroplastic origin of ROS. In addition, oligomerization of RCD1 was addressed in presence of the mitochondrial ETC inhibitors (AA and myx; Figure 4—figure supplement 2B). The latter treatments did not affect the status of RCD1.

Figure 3: Subsection “Evidence for increased electron transfer between chloroplasts and mitochondria in rcd1”: The activation state of NADPH malate dehydrogenase (MDH) is very low, almost unbelievably low (4-8%). Doubling the activity from this very low activation state unlikely is sufficient to explain any changes in rcd1 mutants. Kinetic modelling could help to define whether the observation is significant or not. The legend to the figure is insufficient to understand what is measured. Describe the photosynthetic condition the leaves were exposed to here. The concern with this interpretation is strengthened by the unchanged NADPH/NADP-ratio in non-aqueously isolated chloroplasts. The incubation of the test with and without DTT for 2 h at room temperature is not clear: Was the minus DTT-sample supposed to be the initial activity? During a 2 h incubation with 250 μM DTT, the initially active MDH may inactivate and this may explain the extremely low activity? This ambiguity must be clarified. Possibly the measurements would have to be repeated if they are considered to be central to the paper.

The required information has been added to the Figure 5C legend and to Materials and methods. The activation state of NADPH-MDH is dependent on the growth conditions and light intensity. Low values obtained in this study may be associated with growth conditions (100-120 μmol m-2 s^-1^ at an 8-hour day photoperiod). Most importantly, the NADPH-MDH activation state was different between *rcd1* and Col-0, while the absolute values of this parameter are of secondary importance for the study. However, these values are comparable to those previously reported in tomato (Centeno, D. C. et al., 2011, The Plant Cell).

The comfort of reading is not always optimal at present since the reader must simultaneously consult the text, the legend and the method.

As has been outlined in ##6-8, the manuscript including the figure legends has been profoundly rewritten to facilitate the reading.

The high concentrations of NADH and NADPH in the vacuole appears surprising and an artefact. The description of the method and results may not be complete here. Which other compartment markers were measured and included in the best fit? May be it would be more reliable and convincing to present total leaf nucleotide levels rather than somewhat questionable date on subcellular compartments. The statistically significant difference in vacuolar NADH points to severe problems with this method.

Non-aqueous fractionation does not allow separation of vacuolar and apoplastic compartments, which is likely to be the source of the discrepancy. To clarify this, the figures (Figure 1—figure supplement 4C and Figure 5—figure supplement 2) have been corrected, “vacuole” compartment has been relabelled as “vacuole/ apoplast”. Figure 1—figure supplement 4C legend was updated by adding the following text: “Note that the method does not allow for separation of apoplastic and vacuolar compartments or reliable definition of the mitochondria (Fettke et al., 2005)”.

The discussion is mostly fine, but sometimes quite speculative. Thus the scenario described for ROS-mediated aggregation of RCD1 and subsequent release of RCD1-dependent repression may still be considered as hypothesis. Were the ANAC13/ 17 targets expressed in the lines complemented with the heptuple Cys-to-Ser variant upon oxidative treatment?

The Discussion and the Figure 8 legend have been modified to highlight the hypothetical nature of the proposed regulation. The expression of the ANAC013 /ANAC017-regulaterd genes in the RCD1Δ7Cys-HA line has been assessed (see answer #2). This new data was added as Figure 2—figure supplement 1D.

Reviewer #2:In this study, Shapigzov, Vainonen et al. further explore the role of RCD1 (previously picked up in an ozone sensitivity screen) in regulation of electron transfer and retrograde signalling. […] The point of the experiment shown in Figure 3A may not be very clear to the general reader, and needs better explanation.

The measurement of Fs under combined treatment with MV and SHAM has been described in more detail in the Results and in the Figure 5A legend. In addition, new Figure 1—figure supplement 1A has been introduced to familiarize the reader with the chlorophyll fluorescence parameters addressed in the study. Furthermore, as a parallel approach to the Fs measurement, PSII photochemical quenching was measured as presented in Figure 5—figure supplement 1.

The authors then check NADPH-MDH activity and malate content. Both are higher in rcd1 than in Col-0 after MV treatment. Overall NADPH-MDH capacity was higher in rcd1, suggesting some flux of reductive equivalent to the mitochondria via chloroplastic malate formation.[…] The Sal1 and rcd1 sal1 mutants would need to be analysed in more detail (e.g. Fv/Fm MV analysis) to show PAP is really involved here.

MV tolerance of *sal1* and *rcd1 sal1* mutants has been assessed and presented as Figure 6—figure supplement 2.

Constitutive induction of MDS transcripts (including ANAC013) genes suggests that their main positive transcription factors are activated (ANAC017/013). To address this, the authors made the interesting cross of rcd1 anac017. Indeed, AOX levels and capacity are lower in this double mutant, supporting the role of ANAC017 in MDS upregulation in rcd1. The anac017 mutant was previously shown to be highly MV-sensitive, and ANAC013/ANAC017 overexpression lines are more tolerant (e.g. De Clercq et al., 2013). The observed hypersensitivity of the anac017 mutant in the Fv/Fm MV assay is in line with this. The rcd1 anac017 is more sensitive than rcd1 indicating the NAC MRR pathway plays a partial role in the rcd1 MV phenotype. No description of the visual, developmental and stress (e.g. ozone) phenotype of the rcd1 anac017 mutant is provided, which would be very interesting.

In the *rcd1 anac017* double mutant the curly leaf and the ozone sensitivity phenotypes were not complemented. We believe that demonstrating this data would distract the reader from the storyline and complicate the reading. Our results suggest that this phenotype is largely independent from the organelle interactions (this is addressed in the last paragraph of Discussion and mentioned in Figure 6—figure supplement 1 legend).

The authors then support the previously reported interaction of ANAC013 with RCD1 (Jaspers et al., 2009, O'Shea et al., 2017) using Co-IP strategies, and provide further insight into the interaction with surface plasmon resonance and NMR.

*Finally, the authors show that RCD1 protein level (or at least HA-RCD1) is degraded over time by MV and* H_2_O_2_*treatment. Using non-reducing gels, they show that quickly after these treatments, a more oxidised multimeric form of RCD1 is found, indicating protein aggregation. From my interpretation of Figure 6B and Supp S5B,* H_2_O_2_*makes a predominantly much higher MW aggregate than MV, so this should be discussed.*

The apparent different positions of H_2_O_2_ and MV-related aggregates of RCD1 in the old version of Figure S5B was an artefact caused by the bending of the stacking gel. To avoid discrepancies, this figure has been replaced with the new Figure 4—figure supplement 2B. As it is clear from the new figure, in both treatments the aggregates migrate at a similar height.

*By mutating 7 Cys residues in the RCD1 protein, the authors showed that this aggregation can be largely blocked. I am somewhat confused by the statement on mitochondrial inhibitor AA in the second paragraph of subsection “RCD1 protein is sensitive to organellar ROS” and Figure S5B. To me the total (reducing gel) abundance of RCD1 after AA is much lower than after MV. This seems in contrast to the statement in the text that RCD1 is not a direct target of a mitochondrial signal from AA. This should be clarified. Also* H_2_O_2_*seems much stronger in affecting RCD1 abundance than MV. In line with this, MV is a much weaker inducer of MDS genes than AA or* H_2_O_2_*(Figure 4A), so perhaps the role of RCD1 is of more importance in AA signalling than in MV, but this is under explored.*

In the old Figure S5B antimycin A (AA) was used in the concentration 20 μM, which is much higher than in rest of this study (2.5 uM). The decrease in RCD1-HA abundance was reproducibly observed under 20 μM AA, but not under 2.5 μM AA. However, the data on 20 μM AA is hard to interpret due to possible chloroplastic off-target effects of ≥ 20 μM AA, as described in (Watanabe et al., 2016). Most importantly for this study, 2.5 μm AA (or 2.5 μm myxothiazol) was sufficient for MDS gene induction, however it did not lead to changes in either RCD1-HA abundance or redox states. To show this, the new figures were added to the manuscript (Figure 4—figure supplement 2A, B). The ambiguous results with 20 μM AA were excluded. Together with the fact that AA- or myx-triggered MDS gene induction was also observed in the *rcd1* mutant (Figure 4—figure supplement 1B) this suggested that RCD1 is unlikely part of the AA signalling. The Discussion has been updated to state this more explicitly.

In S5C, I assume the legend is mislabelled, where Col-0 should be the grey line? The identical phenotype of the 7cys RCD1 complementation compared to the normal RCD1 complementation, however questions the physiological importance of the RCD1 redox-induced aggregation, as one might expect then less efficient complementation.

For the clearer demonstration of MV tolerance of RCD1-HA and RCD1Δ7Cys-HA complementation lines, the new datasets have been added to the manuscript (Figure 1—figure supplement 1B, and Figure 2—figure supplement 1C). As can be seen in these figures, at certain low MV concentration some of the RCD1-HA complementation lines are more sensitive to MV than Col-0 (which was visible in the old Figure S5C). The added data shows that mutation of the RCD1 cysteines affects in vivoabundance and stability of the protein. This has been addressed in Results and in Discussion.

Overall, the manuscript contains lots of data, and the experiments seem well performed so I trust their validity. […]To make this more convincing, more direct experiments should be performed. One would for instance expect that an RCD1 Overexpression line (which has been made in the past, Fujibe 2006, Biosci. Biotechnol. Biochem., 70 (8), 1827-1831, and complements the rcd1 MV resistance phenotype) would suppress (AA-induced) MRR to a significant extent. The 7cys RCD1 line would also be of interest in this context, to show that redox state can affect retrograde signalling.

These questions have been addressed. Please, refer to the answers ##1, 2.

A complementation line of mutated RCD1 lacking the clearly defined ANAC interaction domain should not be able to restore wild-type levels of MV susceptibility and MDS upregulation, but I am not aware if one is available. Experiments showing a dynamic release of sequestration/binding of ANACs from RCD1 during stress would be even more convincing, but I agree these are technically very challenging.

This question has been addressed. Please, refer to the answer #3.

Overall, the study provides many new insights and is well written, and soundly presented. To really make a convincing case for their exciting model, some more specific experiments targeting the ANAC/RCD1 regulation would be beneficial.

[Editors' note: the author responses to the re-review follow.]

Reviewer #1:In this revised version the authors addressed many small and large issues raised by the reviewers and editor. However, I feel some of the main concerns have not been addressed appropriately yet.The authors try to introduce a new mechanism of how RCD1 directly binds and inhibits ANAC017/13[…] This interesting model has a number of testable implications, which were suggested after initial review, and were also underlined to be important by the Editor. However some of these have not been addressed directly.1) In comment by eLife Senior Editor: "One important experiment would be to look for suppression of NAC17-dependent retrograde signalling in an RCD overexpression and/or RCD 7cys complemented line[…] The 7cys RCD1 line would also be of interest in this context, to show that redox state can affect retrograde signalling."The authors respond in point 1 that a number of rcd1 RCD1-HA with higher and lower levels of RCD1 (thus probably equivalent to an RCD1 overexpression line, but hard to say without RCD1 specific antibody, and only an anti-HA antibody) have been generated.

In response to these concerns raised by the reviewers, additional data has been added to the study, as follows:

1) Abundance of RCD1 protein in different *rcd1*: RCD1-HA complementation lines in comparison to the wild type has been assessed in Figure 1—figure supplement 1B. Information on the antibody against RCD1 that we have generated has been added to the Materials and methods section.

2) Abundance of *RCD1* mRNA in the *rcd1*: RCD1-HA lines was measured by qPCR and compared to that in the wild type. This has been included as Figure 1—figure supplement 1C. Taken together, the immunoblotting and the qPCR results demonstrate that the *rcd1*: RCD1-HA lines expressing higher levels of RCD1-HA are the true overexpressors of RCD1, as the level of both the RCD1 protein and the *RCD1* transcript exceed those in Col-0 about 10-fold. Moreover, in the *rcd1*: RCD1-Venus lines that we generated, *RCD1* is expressed under *UBIQUITIN-10* promoter. In these lines RCD1 accumulates to amounts even higher than in the shown high-expressor *rcd1*: RCD1-HA lines. However, even in the *pUB10:RCD1* lines we have never observed any signs of restoration of the *rcd1*-like phenotypes. Thus, we believe that the physiological response reported in (Fujibe et al., 2006) is not associated with overexpression of RCD1 protein, but has to do with more complex secondary effects such as silencing of other members of the SRO family. Thus, for the experiments proposed by the reviewers we have used the set of independent *rcd1*: RCD1-HA lines expressing different levels of RCD1-HA (please, see below).

The authors then show the MV-dependent Fv/Fm phenotype is reverting to different extents to Col-0. This is nice, but the key experiment to address the raised question is then missing. When treated with a Mitochondrial Retrograde Regulation inducing agent (classically antimycin A), do these “overexpression” lines show a reduction in ANAC017-dependent MRR gene expression (e.g. Aox1a, UPOX, UGT74E2)? The authors did experiments along these lines but using MV (Figure 2 sup 1D). MV however does not directly induce mitochondrial signalling as it mainly affects chloroplastic ROS, which is supported by the fact that MRR marker genes AOX1a and UPOX are not induced by MV (Figure 2—figure supplement 1D). I don't see why the authors did not do the same experiment using AA, which would address the question, and is relatively easy to perform. In the RCD1d7cys complemented line, which fails to aggregate, such a repression of AA-induction of MRR genes may not occur, and would thus be expected to show more WT-like gene expression responses.

The suggested experiment has been conducted. The new data has been added as Figure 6—figure supplement 2 (induction of MDS genes after 3-hour treatment with 50 μm AA in Col-0, *rcd1*, independent *rcd1*: RCD1-HA and independent *rcd1*: RCD1Δ7Cys-HA lines), and as Figure 8—figure supplement 1 (same treatment performed in the *rcd1-1 anac017* double mutant). As outlined above, *rcd1*: RCD1-HA lines expressing higher levels of RCD1-HA were used as RCD1 overexpressors. In agreement with the reviewer’s prediction, we have observed suppressed induction of the *UPOX* gene in the *rcd1*: RCD1-HA high expressor line and in the *rcd1*: RCD1Δ7Cys-HA lines.

2) A second experiment that was suggested was to use rcd1 mutants complemented with RCD1 lacking the ANAC017 interaction domain (RST domain). In point 3 the authors describe RCD1dRST-HA complementation lines were generated. I assume this is indeed transformation into the rcd1 mutant background, so I suggest these line are called rcd1 RC1dRST-HA in the text and figures (e.g. Figure 7B-C), same for rcd1 RCD1-HA. This to avoid confusion that they are in the Col-0 background, which would lead to a different interpretation.

The figures and the text have been modified accordingly.

The results in Figure 7B and C are indeed in line with the fact that the RST1 domain is important for RCD1 function in this context. As RCD1 is known to interact with a wide range of transcription factors, so it is not direct evidence for a role via interaction with ANAC017/13, but overall that is a nice result.[…] 5) I still think it is important to show the visual phenotype of the rcd1 anac017 mutant plants, even if they don't seem to revert the rcd1 ozone resistance phenotype, as is partially observed for the MV Fv/Fm phenotype. This makes some sense, as anac017 is not known to be involved in ozone tolerance.

Since previous version of the manuscript, we have discovered growth conditions under which we see partial complementation of *rcd1* habitus phenotype in the *rcd1 anac017* double mutant. This was included as the new main figure data panel. To make the results presented in the crowded former Figure 7 more visual, this figure was split in two: Figure 7 and Figure 8.

*6) I don't really understand the author's reasoning in point 37 of their reply letter. The authors say there is no effect of AA on RCD1-HA redox state, but Figure 4—figure supplement 2B to me shows induction of higher order complexes by AA and myx of the same size as MV in light or* H_2_O_2_*, albeit weaker. 2.5 μm is a relatively low concentration, as 50 μm AA has been used most extensively throughout the literature. They also say that because MDS gene induction (by which they mean AOX protein level changes using Western blots) by AA is still occurring in rcd1, this leads them to conclude that RCD1 is unlikely part of AA signalling. As RCD1 is suggested to suppress ANAC017 signalling, an rcd1 KO should still show AA gene induction, as the potential repressor is now inactive. An RCD1 OX line may display reduction in AA signalling, but this question has still not been addressed, see Point 1.*

As stated above, transcriptional response to higher concentration of AA (50 uM) has been addressed and included in the manuscript. In our opinion, the experiment proposed by the reviewers has significantly added to this manuscript. In particular, it showed that *rcd1* mutant was not affected in the initiation of MDS signalling, that the overexpressor and the RCD1Δ7Cys lines had this signalling somewhat suppressed, and that ANAC017 played decisive role in the initiation of the MDS signal both in Col-0 and in *rcd1*.

At the same time, we believe that the in vivo studies of RCD1 under 50 μm AA should be interpreted with caution, because (i) RCD1 may be sensitive to redox states of various subcellular origin, and (ii) high concentrations of AA are known to inhibit chloroplast cyclic electron flow (on which account a cautionary note has been included in the legend of Figure 6—figure supplement 2). This makes it harder to interpret the effects of high concentrations of AA on RCD1 biochemistry and redox states. Nevertheless, we agree with the reviewers that the possibility of RCD1 acting as a redox sensor of the mitochondrial MDS signalling has not been ruled out in this study. To account for this uncertainty, the Discussion text has been modified accordingly. (“In addition, little change was observed (in RCD1 abundance and redox state) with the mitochondrial complex III inhibitors AA or myx (Figure 4—figure supplement 2A, B). Together with the fact that MDS induction was not compromised in the *rcd1* mutant (Figure 4—figure supplement 1 and Figure 6—figure supplement 2), this suggests that RCD1 may primarily function as a redox sensor of chloroplastic, rather than mitochondrial, ROS/ redox signaling.”)

Reviewer #2:[…] In the following you find the combined list of minor and very few more critical points.Introduction section, first paragraph: “… transport of carbon rich metabolites…” It is not really clear what is meant? Carbon rich might be a fatty acid? energy-rich like dihydroxyacetone phosphate or malate?

The text has been clarified by changing it to “…transport of photoassimilate-derived carbon rich metabolites from chloroplasts to mitochondria.” We believe that such wording would be the most accurate in describing compounds, which may undergo conversion after their export from the plastid and before import into the mitochondria.

Introduction section, second paragraph: “… leakage of electrons…”; is this still an adequate description of the process? Our concept of ROS signaling more and more assumes that ROS are generated to enable regulation.

The wording “occasional leakage” has been replaced with “transfer”.

Introduction section, fourth paragraph: “… provides positive feedback regulation”; please be more precise and give the direction:.… provides positive feedback and thus enhancement of the signal.

The text has been modified accordingly.

Subsection “Mitochondrial respiration is altered in rcd1”, opening sentence: I would omit the U here, since it is confusing for the non-expert, and in the legend and Materials and methods it will be mentioned. But in fact the uniformity of label is not explained, is it?

We assume that the fact that glucose was uniformly labelled is essential for correct interpretation of the results. Thus, it was kept in the text. Definition of the abbreviation has been added in the first use, to facilitate readability.

Subsection “Retrograde signaling from both chloroplasts and mitochondria is altered in rcd1”: msh1, recA3 (add comma)

*msh1 recA3* is a double mutant and according to Arabidopsis gene nomenclature standard it should be presented as it was in previous submission.

Discussion third paragraph: “suggests that RCD1 is not a redox sensor of mitochondrial ROS/ redox signaling” still sounds like an overinterpretation of the data and should be toned down. “unlikely is…” In fact in context with the subsequent discussion, it is not really clear what this statement means. If mitochondrial alterations in metabolism cause redox changes in the nucleus/cytosol similar to those of the chloroplast, then the conformational changes should be the same. The power of mitochondrial redox/ROS signals may just be less than chloroplast redox/ROS signals.

The text in the Discussion has been modified accordingly. Please, see the reply above.

Subsection “RCD1 regulates stress responses and cell fate”, first paragraph: Rap2.4a

The phrase in question referred to the families of the transcription factors, not to their individual members.